# CLEPS 1.0: A new protocol for cloud aqueous phase oxidation of VOC mechanisms

Camille Mouchel-Vallon[1,a], Laurent Deguillaume[1], Anne Monod[2], Hélène Perroux[1], Clémence Rose[1], Giovanni Ghigo[3], Yoann Long[1,b], Maud Leriche[4], Bernard Aumont[5], Luc Patryl[6], Patrick Armand[6], Nadine Chaumerliac[1]

[1]Université Clermont Auvergne, CNRS Laboratoire de Météorologie Physique, F-63000 Clermont-Ferrand, France
[2]Aix Marseille Université, CNRS, LCE UMR 7376, 13331, Marseille, France
[3]Dipartimento di Chimica, Universita di Torino, V. Giuria 7, I-10125 Torino, Italy
[4]Université de Toulouse, UPS, CNRS, Laboratoire d'Aérologie, 31400 Toulouse, France
[5]LISA UMR CNRS 7583, Université Paris Est Créteil et Université Paris Diderot, Paris, France
[6]CEA, DAM, DIF, F-91297 Arpajon, France
[a]Now at: Wolfson Atmospheric Chemistry Laboratories, Department of Chemistry, University of York, Heslington, York, YO10 5DD, United Kingdom
[b]Now at: Ramboll Environ, 155 rue Louis de Broglie, 13100 Aix-en-Provence, France

*Correspondence to*: L. Deguillaume (l.deguillaume@opgc.univ-bpclermont.fr), C. Mouchel-Vallon (camille.mouchel-vallon@york.ac.uk)

**Abstract.** Organic compounds of both anthropogenic and natural origin are ubiquitous in the multiphase atmospheric medium. Their transformation in the atmosphere affects air quality and the global climate. Modelling provides a useful tool to investigate the chemistry of organic compounds in the tropospheric multiphase system. While several comprehensive explicit mechanisms exist in the gas phase, explicit mechanisms are much more limited in the aqueous phase.

Recently, new empirical methods have been developed to estimate HO˙ rate constants in the aqueous phase: structure-activity relationships (SARs) provide global rate constants and branching ratios for HO˙ abstraction from and addition to atmospheric organic compounds. Based on these SARs, a new detailed aqueous-phase mechanism, named the cloud explicit physico-chemical scheme (CLEPS 1.0), to describe the oxidation of hydrosoluble organic compounds resulting from isoprene oxidation is proposed. In this paper, a protocol based on reviewed experimental data and evaluated prediction methods is described in detail. The current version of the mechanism includes approximately 850 aqueous reactions and 465 equilibria. Inorganic reactivity is described for 67 chemical species (*e.g.*, transition metal ions, $H_xO_y$, sulphur species, nitrogen species, and chlorine). For organic compounds, 87 chemical species are considered in the mechanism, corresponding to 657 chemical forms that are individually followed (*e.g.*, hydrated forms, anionic forms). This new aqueous-phase mechanism is coupled with the detailed gas phase mechanism MCM v3.3.1 through mass transfer parameterization for the exchange between the gas phase and aqueous phase. The GROMHE SAR (GROup contribution Method for Henry's law Estimate) enables the evaluation of the Henry's law constants for undocumented organic compounds. The resulting multiphase mechanism is implemented in a model based on the Dynamically Simple Model for Atmospheric Chemical Complexity (DSMACC) using the Kinetic PreProcessor (KPP). This model allows simulation of the time evolution of the concentrations of each individual chemical species in addition to detailed time-resolved flux analyses. The variable photolysis rates in both phases is calculated using the TUV 4.5 radiative transfer model.

To test our chemical mechanism, an idealized cloud event with fixed microphysical cloud parameters is simulated. The simulation is performed for a low-$NO_x$ situation. The results indicate the formation of oxidized mono- and diacids in the aqueous phase, as well as a significant influence on the gas phase chemistry and composition. For this particular simulation, the aqueous phase mechanism is responsible for the efficient fragmentation and functionalization of organic compounds. This new cloud chemistry model allows for the analysis of individual aqueous sub systems and can be used to analyze the results from cloud chamber experiments and field campaigns.

## 1 Introduction

Clouds favour chemical reactions that would not occur in the gas phase or at a rate much slower than in the aqueous phase (Epstein and Nizkorodov, 2012; Herrmann, 2003; Herrmann et al., 2015). Reactivity in clouds is due to (1) highly enhanced photochemical processes in cloud droplets; (2) faster aqueous phase reactions than in clear sky, some of which do not occur in the gas phase, especially those involving ions and hydration; (3) possible interactions between the aqueous phase and particulate phase. Clouds can also be responsible for secondary organic aerosol (SOA) formation and ageing. However, aqueous phase processes suffer from large uncertainties. Blando and Turpin (2000) first proposed clouds as a source of SOAs. Recent field measurements (Kaul et al., 2011; Lee et al., 2012, 2011), experimental work (Brégonzio-Rozier et al., 2016) and modelling studies (Ervens, 2015; Ervens et al., 2011) have shown that aqueous phase processes could lead to SOA formation on the same order of magnitude as gas phase processes. The contribution of cloud and fog processes to SOA formation is firstly indirect, through the effects of cloud chemistry on the oxidant budget. Gas phase reactivity of volatile organic compounds (VOC) is controlled by daytime $HO^{\bullet}$ oxidation, and it has been shown that phase partitioning of its precursors in clouds plays a significant role in the budget of this oxidant (Herrmann et al., 2015). Secondly, cloud and fog processes act directly on SOA sinks and sources. Dissolution and processing of organic vapour in the aqueous phase can lead to the formation and destruction of SOA precursors through accretion (*i.e.,* carbon-carbon bond formation) and oxidation processes. These processes may compete in the aqueous phase (Kirkland et al., 2013; Renard et al., 2015), simultaneously acting as the source (through oligomerization and functionalization reactions) and sink (through fragmentation reactions) of SOAs. To elucidate the contribution of accretion and oxidation to the budget of SOA precursors, most recent studies on the topic have focused on the modelling and measuring of accretion processes (Ervens et al., 2015). However, equivalent knowledge of aqueous oxidation is needed because oxidation processes may control the availability of organic compounds and radicals to form accretion products. In this work, we therefore focus on aqueous oxidation processes, especially the competition between fragmentation and functionalization.

Competition between fragmentation and functionalization processes has been identified as a major factor in the production of SOA in the gas phase (Donahue et al., 2012; Jimenez et al., 2009). To better represent these processes in clouds, detailed multiphase mechanisms are needed. Such mechanisms are available for gas phase chemistry (Aumont et al., 2005; Jenkin et al., 1997), whereas very few exist for the aqueous phase. Among the aqueous phase mechanisms, the most cited is Chemical Aqueous Phase Radical Mechanism version 3.0 (CAPRAM 3.0). CAPRAM 3.0 represents one oxidation pathway for each considered species (Herrmann et al., 2005; Tilgner and Herrmann, 2010; Whalley et al., 2015), even if simple organic molecules react at several distinct oxidation sites at comparable rates. As a consequence, there is a loss of information in such mechanism that does not accurately represent both fragmentation and functionalization. To fill this gap, a new protocol is proposed to create an explicit aqueous phase oxidation mechanism to accurately represent the various oxidation pathways of organic matter: the Cloud Explicit Physicochemical Scheme (CLEPS 1.0).

This protocol is developed under low-$NO_x$ (< 1 ppbv) and dilute conditions and uses recently available laboratory data and empirical estimation methods. The mechanism is implemented in a box model including the gas phase chemistry and kinetic mass transfer of soluble species between the gas phase and cloud droplets. The box model is tested in an ideal cloud situation.

## 2 Overview

The aqueous phase oxidation mechanism originally relied on inorganic chemistry (see Deguillaume et al., 2004; Leriche et al., 2007) and the oxidation of several organic $C_1$ and $C_2$ species, including photo-oxidation of iron complexes with oxalic acid (Long et al., 2013). The inorganic processes taken into account in CLEPS are described in Deguillaume et al. (2004) and Long et al. (2013). This inorganic mechanism simulates the redox processes involved in the evolution of $H_xO_y$, sulphur,

nitrogen, halogen (Leriche et al., 2000, 2003) and transition metal ions (TMIs; Deguillaume et al., 2004). A special emphasis is given to the latter, as the speciation of TMIs is believed to drive the evolution of aqueous phase oxidants ($H_2O_2$, $HO^•$, $HO_2^•/O_2^{•-}$; Deguillaume et al., 2005) (see R14 to R95 in the mechanism tables). This inorganic part of the mechanism is very similar to other available aqueous phase mechanisms. For example, the chemistry of transition metal ions includes Fe, Cu and Mn redox cycles also considered in the CAPRAM 2.4 mechanism (Ervens et al., 2003). Sulphur, nitrogen and $H_xO_y$ systems are also relatively well documented (Herrmann et al., 2010). The CLEPS and CAPRAM mechanism generally consider similar reactivity for these chemical subsystems.

In the present study, the CLEPS mechanism is extended to the oxidation of $C_{1-4}$ precursors and follows the protocol described in detail in sections 3 and 4. Although isoprene is not significantly dissolved in the atmospheric aqueous phase, its oxidation products are considered. For instance methylglyoxal (MGLY), glyoxal (GLY), acrolein (ACR), methacrolein (MACR) and methylvinylketone (MVK) are significantly soluble (Henry's law constant $> 10^3$ M atm$^{-1}$) and/or highly reactive in the aqueous phase (Ervens and Volkamer, 2010; Lim et al., 2010, 2013; Liu et al., 2009, 2012). Ervens et al. (2015) however argued in a modeling study that MVK and MACR solubility could be decreased by salting out effects, reducing their contributions to aqueous reactivity and SOA formation.

For each species and its oxidation products, the CLEPS mechanism describes the oxidation of $HO^•$ (section 3.3) and $NO_3^•$ (section 3.4) and the explicit evolution of the produced peroxyl radicals (section 4). When rate constants are available, the reactivity of organic compounds with other oxidants ($SO_4^•$, $Cl_2^•$,…) is also addressed (section 3.5). Hydration (section 3.1) and dissociation (section 3.2) equilibria are, respectively, considered for carbonyl and carboxylic functions.

Moreover, recent developments in empirical estimates of kinetic and thermodynamic parameters (*e.g.*, rate constants, Henry's law constants) for aqueous phase chemistry (Doussin and Monod, 2013; Minakata et al., 2009; Monod and Doussin, 2008; Raventos-Duran et al., 2010) are included in the CLEPS mechanism. These structure activity relationships (SARs) are based on experimental data and rely on robust hypotheses about the rate constants (section 3.3.2) and equilibrium constants (sections 3.1 and 3.2) of species that are not documented in the literature. For instance, SARs can provide estimations of the branching ratios between the different oxidation pathways with $HO^•$ radicals (Doussin and Monod, 2013; Minakata et al., 2009; section 3.3.3).

The mechanism currently includes 850 aqueous reactions and 465 equilibria. Inorganic reactivity is described for 67 chemical species (*e.g.*, TMIs, $H_xO_y$, sulphur species, nitrogen species and chlorine). For organic compounds, 87 chemical species are considered in the mechanism corresponding to 657 chemical forms (*e.g.*, hydrated forms, anionic forms). The mechanism tables are available in the supplementary data.

## 3 Stable organic species

### 3.1 Hydration equilibria

Carbonyls, *i.e.,* aldehydes and less likely ketones, may undergo hydration leading to the formation of a gem-diol form:

**R.1** $\quad >CO + H_2O \overset{K_{hyd}}{\longleftrightarrow} >C(OH)(OH)$

$K_{hyd}$ [dimensionless] is the hydration constant and is defined as:

**Eq.1** $\quad K_{hyd} = \frac{>C(OH)(OH)}{>CO}$

There are 30 carbonyl species in the mechanism. Most of the $C_{1-2}$ species are well known and data are available in the literature. However, there is a lack of data for $C_{3-4}$ species, and empirical estimates must be performed.

To the best of our knowledge, there is only one SAR available to estimate hydration constants (Raventos-Duran et al., 2010); it is provided by the GROMHE (GROup contribution Method for Henry's law Estimate) SAR for Henry's law constants.

This SAR is based on five descriptors and is optimized on a dataset comprising 61 species. Raventos-Duran et al. (2010) defined a global descriptor, *tdescriptor*, to represent functional group interactions with the sum of the so-called sigma Taft values ($\sigma^*$, *e.g.,* Perrin et al., 1981). Similarly, *hdescriptor* is a global descriptor representing the inductive effect of functional groups attached to an aromatic ring through the sum of the meta-, para- and ortho-Hammett sigma values ($\sigma_m$, $\sigma_p$ and $\sigma_o$, respectively; *e.g.,* Perrin et al., 1981; for more details see Raventos-Duran et al., 2010). It is applied for all stable carbonyl species when a measured value is not available. However, this method was originally developed only for stable non-ionic species.

In the present study, the SAR is extended to anionic species. The descriptors have been optimized to include the Taft and Hammett sigma values for the carboxylate moieties ($\sigma^*$(-CO(O$^-$)) = -1.06, $\sigma_m$(-CO(O$^-$)) = 0.09, $\sigma_p$(-CO(O$^-$)) = -0.05 and $\sigma_o$(-CO(O$^-$)) = -0.91; Perrin et al., 1981). The database from Raventos-Duran et al. (2010) has been extended to carboxylate species with measured values available in the literature (**Table 1**). Following the same method as Raventos-Duran et al. (2010), multiple linear regression optimization is performed by minimizing the sum of squared errors (SSE):

**Eq.2** $$SSE = \sum_{i=1}^{n}\left(\log K_{hyd,est} - \log K_{hyd,exp}\right)^2$$

where $n$ is the number of experimental values in the database (n=65), including both the values compiled and taken into account by Raventos-Duran et al. (2010) and the new carboxylate values. **Figure 1** shows both the previous and the updated values of the descriptors and the performance of the SAR. The new optimization for carboxylate compounds modifies the descriptors by at most 50% for the intercept. The other descriptors vary from 1% (*hdescriptor*) to 18% (*ketone flag*) of their initial values. There is greater uncertainty associated with this new optimization; the root mean square error (RMSE = $\sqrt{\frac{1}{n}\sum_{i=1}^{n}\left(\log K_{hyd,est} - \log K_{hyd,exp}\right)^2}$) is 0.61 log units, which is higher than the RMSE = 0.47 log units given in Raventos-Duran et al. (2010). However, the new optimization is still able to estimate $K_{hyd}$ within a factor of 4 (3 in Raventos-Duran et al., 2010).

Hydration data are not available in the literature for the peroxyl (RO$_2{}^\bullet$) and acylperoxyl (RC(=O)(O$_2{}^\bullet$)) radicals. However, there is no reason to ignore the hydration of these radicals. As a first approach, when data are not available, we assign to a given RO$_2{}^\bullet$ the hydration constant of its parent species. For example, for the radicals derived from glycolaldehyde, we have:

**Eq.3** $$K_{hyd}\left(CH_2(OH)CO(OO\cdot)\right) = K_{hyd}(CH(OH)(OO\cdot)CHO) = K_{hyd}(CH_2(OH)CHO) = 10$$

The lack of experimental data does not allow validation of this hypothesis and excludes further extension of the Raventos-Duran et al. (2010) SAR. This approximation is likely valid when there is no or weak interactions between the peroxyl and carbonyl functions, *i.e.,* when these functions are separated by several carbon atoms. This is also applied to short-chain hydrocarbons and acylperoxyl radicals since it is the only way to counteract the lack of experimental data. When new data become available, they can be easily implemented to replace our hypothesis.

In this protocol, the hydration equilibria are differently considered from what is done in CAPRAM 3.0 (Herrmann et al., 2005; Tilgner et al., 2013). In CAPRAM, hydration equilibria constants are documented as well as back and forward reactions rates when available. When no data are available, hydration constants from similar species are used.

### 3.2 Acid dissociation equilibria

To represent acid dissociation, the acidity constant $K_A$ [M] is needed:

**R.2** $$-CO(OH) \overset{K_A}{\leftrightarrow} -CO(O^-) + H^+$$

The acidity constant $K_A$ is defined by:

**Eq.4** $\quad K_A = \dfrac{[\text{-CO(O}^-)][\text{H}^+]}{[\text{-CO(OH)}]}$

In general, the $K_A$ values are well documented for short-chain organic compounds ($C_{1-2}$). In the mechanism, each stable acid with one or two carbon atoms can be documented using acidity constant from the literature. This is not the case for $C_3$ and $C_4$ species, especially multifunctional species. In the whole mechanism, there are 38 organic acids, 7 of which have a documented $K_A$ value (see Equilibria Tables in the mechanism tables). Like the hydration constants, the acidity constants must therefore be empirically estimated. To obtain the estimates, we use a similarity criterion: if there are no data available in the literature for a given species, the acidity constant from the closest documented species with the same organic function in the α position from the carboxylic acid function is chosen. For example, the acidity constant for pyruvic acid (Lide and Frederikse, 1995) is attributed to 3-oxopyruvic acid because they both carry a ketone function next to the carboxylic acid function, *i.e.*:

**Eq.5** $\quad K_A(\text{CHOC(=O)CO(OH)}) = K_A(\text{CH}_3\text{C(=O)CO(OH)}) = 10^{-2.4}$

Perrin et al. (1981) showed that the $pK_A$ values of aliphatic organic species are mostly influenced by the inductive effect of the organic function closest to the acidic function. Therefore, this first hypothesis should provide a good estimation of undetermined acidity constants.

Following the hydration constant treatment, the acidity constants for peroxyl radicals are initially taken from their parent species when experimental data are not documented. This assumption can be questioned but very few measurements suggest that peroxyl radicals are more acidic than their parent species. Schuchmann et al. (1989) showed that the acetic acid peroxyl radical ($\text{CH}_2(\text{OO}^\bullet)\text{CO(OH)}$) has a pKa = 2.10, whereas acetic acid has a pKa = 4.76, and they observed the same trend for the malonic acid peroxyl radical, which has a second pKa close to 3 compared to the malonic acid second pKa = 5.7. Therefore, the hydration constants from Schuchmann et al. (1989) are used in the mechanism, and the estimated hydration constants can be directly substituted by laboratory data when the data become available.

Our acidity constant estimates are similar to what is proposed in the other explicit aqueous phase chemistry mechanism such as CAPRAM. We systematized the procedure used in CAPRAM where unknown acidity constants are estimated from similar species.

### 3.3 Reaction with HO$^\bullet$

### 3.3.1 Mechanism

For aliphatic organic compounds, HO$^\bullet$ reactivity proceeds by H-abstraction, yielding an alkyl radical following what can occur in the gas phase (Herrmann et al., 2010, 2015):

**R.3** $\quad \text{RH+HO}\cdot\longrightarrow\text{R}\cdot + \text{H}_2\text{O}$

If the compound bears a C=C double bond, the addition is favoured:

**R.4** $\quad \text{>C=C<+HO}\cdot\longrightarrow\text{>C(OH)-C}\cdot\text{<}$

In addition to the gas phase like pathways described above, it may be possible for HO$^\bullet$ to undergo electron transfer in the presence of anions, especially carboxylate compounds (von Sonntag and Schuchmann, 1997):

**R.5** $\quad \text{-CO}\left(\text{O}^-\right)+\text{HO}\cdot\longrightarrow\text{-CO(O}\cdot) + \text{OH}^-$

### 3.3.2 Rate constants

When rate constants of organic compounds reactions with HO$^{\bullet}$ are available (see the review from Herrmann et al., 2010), they are used in the mechanism. In the CLEPS mechanism, for a total of 343 reactions with HO$^{\bullet}$, only 43 kinetic constants are available in the literature. Empirical estimates are thus required in most cases. The estimates are obtained using the recent developments of SARs for the HO$^{\bullet}$ rate. Doussin and Monod (2013) described the extension of a SAR previously published in Monod and Doussin (2008) and Monod et al. (2005). This SAR provides a way to estimate the H-abstraction rate constants for dissolved linear or cyclic alkanes, alcohols, carbonyls, carboxylic acids and carboxylates. This method includes descriptors that consider the effect of functional groups in the α- and β-positions of the abstracted hydrogen atom. For each considered organic moiety, Doussin and Monod (2013) optimized the α- and β-substitution factors. All estimates in the frame of the CLEPS mechanism are within the domain of validity of the Doussin and Monod (2013) SAR.

In the present study, the SAR was modified to account for the electron transfer on carboxylate compounds (R.5). The relevance of this process was discussed by Doussin and Monod (2013). They found an electron transfer rate constant for α-carbonyl carboxylate anions of $k(-C(=O)CO(O^-)) = 2.1 \times 10^8$ M$^{-1}$ s$^{-1}$, but it was not included in the original SAR due to the limited amount of experimental data. Their analysis was restricted to the α-carbonyl bases (especially pyruvate and ketomalonate ions, for which electron transfer is dominant), due to the lack of abstractable H-atoms. However, other carboxylate ions could undergo this type of reaction, even if electron transfer is of minor importance because of the faster H-abstraction reactions. Therefore, in the present study, the SAR from Doussin and Monod (2013) has been modified to include the partial rate constant $k(-CO(O^-)) = 2.1 \times 10^8$ M$^{-1}$ s$^{-1}$ for each possible electron transfer reaction. This partial rate constant is affected by the α- and β-substitution factors in the same way as the original abstraction constants.

For all unsaturated species in the mechanism (*i.e.,* methylvinylketone - MVK, methacrolein - MACR, hydroxyl-methylvinylketone - MVKOH and hydroxyl-methacrolein - MACROH), the addition reactions rates have been evaluated following the literature and similarity criteria. For further developments involving unknown unsaturated compounds, the SAR from Minakata et al. (2009) should be used because it is the only method that can estimate partial addition rate constants on double bonds.

### 3.3.3 Branching ratios

Branching ratios are required to identify the most probable oxidation products. In previous mechanisms, the most labile H-atom was empirically identified (*e.g.*, using Bond Dissociation Energy estimations), and the reaction was assumed to proceed exclusively *via* this H-abstraction pathway (Ervens et al., 2003; Tilgner and Herrmann, 2010). This was the only hypothesis that could be considered because experimental data on the branching ratios in aqueous phase HO$^{\bullet}$ reactions are extremely scarce. The recent introduction of group-contribution-based SAR (Doussin and Monod, 2013; Minakata et al., 2009) allows estimation of the contribution of each pathway to the global reactivity of each species. In our mechanism, for a given species the global reactivity rate is either provided by the literature or by SAR, but the branching ratios are always obtained from SAR estimates.

Furthermore, for simplicity, a reduction hypothesis was considered in the mechanism for each stable species because explicitly writing all possible reactions would yield a huge number of chemical species. For example, Aumont et al. (2005) showed that the number of species formed in the gas phase for such explicit schemes increases exponentially with the size of the carbon skeleton of the parent species. One can assume based on Aumont et al. (2005) that starting from a C$_4$ precursor in the aqueous phase, the mechanism would require approximately $10^3$ distinct species involved in approximately $10^4$ reactions. Such a large set of species excludes the development of an aqueous phase oxidation mechanism by hand. A simple reduction scheme was therefore applied to mitigate this problem and an example is shown in **Table 2** for three selected species. Doussin and Monod (2013) SAR was applied to estimate the contribution of each possible pathway and to maintain at least

75% of the total reactivity. After the reduction is applied, the branching ratios are recalculated to maintain the global oxidation rate constant.

This empirical reduction scheme helps to limit the number of species and reactions (657 different chemical forms representing 87 species reacting in 673 oxidation reactions). This new aqueous phase mechanism then allows consideration of the most probable H-abstraction reaction. For example, in **Table 2** for the hydrated glycolaldehyde, the final reactivity in our mechanism is equally distributed between three HO• attack sites and yields 33% glyoxylic acid, 28% glyoxal, 39% formic acid and formaldehyde. This result can be compared with the mechanism of Ervens et al. (2004), which leads to 100% glyoxylic acid since it only considers the abstraction of the aldehydic hydrogen.

**Table 2** shows that the Doussin and Monod (2013) SAR estimates often lead to a significant abstraction of the hydrogen atom bonded to the oxygen atom in the hydroxyl moiety. This mechanism has never been addressed in an atmospheric chemical scheme. This reactivity of the alcohol function towards the HO· radical has been experimentally demonstrated by Asmus et al. (1973) for methanol, ethanol, tert-butanol and polyols. To determine whether this reactivity could be extended to all the considered alcohol functions in our mechanism, we investigated whether the process appears thermodynamically feasible by calculating the relative reaction free energies (Gibbs energies) using the Density Functional Theory (DFT, see details and references in the Supplementary Material **SM1**). The thermodynamic values for the reaction enthalpies and Gibbs energies are calculated for the H-abstraction by the HO• radical from C and O atoms in the following molecules: acetaldehyde, propionaldehyde, glycolaldehyde, glyoxal, methylglyoxal, L-lactic acid and L-lactate. H-abstraction from hydrated aldehydes is also included in this study. The results (see Supplementary Material **SM2**) show that H-abstraction is thermodynamically favourable: the reaction enthalpies range from -14 to -47 kcal mol$^{-1}$ and the corresponding free energies range from -13 to -47 kcal mol$^{-1}$.

The SAR from Doussin and Monod (2013) was recently published. Therefore, Herrmann et al. (2005) and Tilgner et al. (2013) could not use this method to estimate rate constants and branching ratios. Instead, they rely on similarities when data are not available: for instance, they assume that the HO• addition rate constant on 2,3-dihydroxy-4-oxobutanoic acid is the same as maleic acid. In most cases, they assume that the reaction will proceed through the identified most probable pathway. In some cases, like for 2,4-butanedione, they attribute branching ratios from the equivalent measured gas phase reaction.

### 3.4 Reaction with NO$_3$•

NO$_3$• is the main night-time oxidant in the gas phase. Although it plays a minor role under low-NO$_x$ conditions, NO$_3$• chemistry has been taken into account in the protocol to make it versatile in the future. Previous modelling studies of aqueous-phase reactivity expect the same characteristics for dissolved NO$_3$• radicals (Tilgner et al., 2010). For this reason, we represent NO$_3$• oxidation for each stable species in the mechanism.

### 3.4.1 Mechanisms

The mechanism of NO$_3$• oxidation is similar to that of HO• oxidation: the reactivity mainly proceeds via H-abstraction of a labile hydrogen atom to form an alkyl radical and nitric acid:

**R.6**     $RH + NO_3 \cdot \longrightarrow R\cdot + HNO_3$

In this version of the mechanism, the addition of an NO$_3$• radical to the C=C double bond is not considered since insufficient data are available for these reactions and for the fate of the resulting organonitrate peroxyl radicals. However, organonitrate compounds were recently identified in the ambient aerosol (Garnes and Allen, 2002; Lee et al., 2016; Rollins et al., 2012). In addition to local emission sources, organonitrate compounds originate from the gas phase chemistry of VOCs under high NO$_x$ conditions (Darer et al., 2011; Farmer et al., 2010; Heald et al., 2010; Paulot et al., 2009; Perring et

al., 2013) followed by phase transfer and aqueous processing to the deliquescent aerosol and cloud aerosol phase (Nguyen et al., 2011) to form nitrates, alcohols or organosulphates, which contribute to SOA. In this study, we restricted the simulations to low-$NO_x$ conditions to reduce the potential importance of organonitrate chemistry, which requires further experimental and modelling developments.

### 3.4.2 Rate constants

Data concerning $NO_3^{\bullet}$ reactions rates are available in the literature, mostly for $C_1$ and $C_2$ species (Exner et al., 1994; Gaillard de Sémainville et al., 2007, 2010; Herrmann et al., 2010, 2015). Again, empirical estimates are required to describe the oxidation by $NO_3^{\bullet}$ radicals when the data are unavailable.

For most $C_{2-4}$ species, we estimate the $NO_3^{\bullet}$ rate constants using the similarity criteria. When an estimate is needed, we use the reaction rate of a similar documented species. The primary focus is on the nature, number, and relative position of the organic functions. For example, the reaction rate of 3-hydroxypropionic acid with $NO_3^{\bullet}$ is estimated to be the same as the reaction rate of lactic acid because both are $C_3$ species with a carboxylic acid and a hydroxyl function:

**Eq.6**     $k(CH_2(OH)CH_2CO(OH) + NO_3 \cdot) = k(CH_3CH(OH)CO(OH) + NO_3 \cdot) = 2.1 \times 10^6 \ M^{-1}s^{-1}$

(De Semainville et al., 2007)

### 3.4.3 Branching ratios

The branching ratios for $NO_3^{\bullet}$ oxidation are not available in the literature. As a first approach, we use branching ratios for the $NO_3^{\bullet}$ reactions that are identical to those estimated for $HO^{\bullet}$ reactions because $NO_3^{\bullet}$ H-abstraction proceeds following the same elementary mechanism as $HO^{\bullet}$ H-abstraction. $NO_3^{\bullet}$ radicals should be more sensitive to steric hindrance than $HO^{\bullet}$ radicals. However, without experimental evidence supporting this assumption, in our mechanism, $NO_3^{\bullet}$ radicals are unable to react *via* electron transfer and to abstract hydrogen atoms from -OH moieties. In these cases, electron transfer and -OH hydrogen abstraction are not included in the list of available $NO_3^{\bullet}$ reactions, and the remaining pathways are rescaled to 100%. Therefore, for these types of reactions, the branching ratios for $NO_3^{\bullet}$ oxidation may differ from those for oxidation by $HO^{\bullet}$.

Our approach for reactions with nitrate radicals is different than what has been proposed in previous aqueous phase mechanism, especially for estimating the branching ratios. We use similarity criteria to estimate the reaction rates whereas Herrmann et al. (2005) estimate these values using their own SAR (Herrmann and Zellner, 1998). They assume that the H-abstraction only occurs at the weakest C-H bond, which is determined from bond dissociation energies empirical estimates.

### 3.5 Reaction with other oxidants

Reactions rates with radicals other than $HO^{\bullet}$ or $NO_3^{\bullet}$ are available in the literature (Herrmann et al., 2015; Zellner et al., 1995). They mainly concern reactions of $C_{1-2}$ species with $Cl_2^{\bullet-}$, $CO_3^{\bullet-}$, $FeO^{2+}$ and $SO_4^{\bullet-}$. These reactions are included in the mechanism, and the branching ratios are based on the $HO^{\bullet}$ reaction branching ratios. However, reactions with these radicals are not included when information in the literature is unavailable, which would require further extension of the CLEPS mechanism when applied to polluted (relatively rich in $SO_4^{\bullet-}$ radicals) or marine (relatively rich in $Cl_2^{\bullet-}$ and other halogenated radicals) environments.

Since our mechanism is developed for low-$NO_x$ conditions, the reactivity of selected oxygenated organic species with $H_2O_2$ and $O_3$, which was recently studied by Schöne and Herrmann (2014), is included in the mechanism (see reactions R586 and R590 in the mechanism tables). Although these reaction rates are in the range of $10^{-1}$ to $10^1 \ M^{-1} \ s^{-1}$, their impact is non-negligible under specific conditions, especially under low-$NO_x$ conditions. Furthermore, reactions of $C_{1-2}$ hydroperoxide

compounds in Fenton-like reactions with $Fe^{2+}$ have been studied by Chevallier et al. (2004) and are included in the mechanism (see reactions R238 and R330 in the mechanism tables).

## 3.6 Photolysis

Most of the species considered in the mechanism are oxygenated and are likely to bear chromophore functional groups. To calculate the photolysis reaction rate, the polychromatic absorption cross-sections and quantum yields must be known. Again, the literature data concerning these subjects are scarce. Photolysis data are available for a few chromophore species: $H_2O_2$ (Graedel and Weschler, 1981; Zellner et al., 1990), carboxylate-iron(III) complexes (Faust and Zepp, 1993; Long et al., 2013; Weller et al., 2013a, 2013b), and pyruvic acid (Reed Harris et al., 2014). Absorption cross-section and quantum yield data (preferably wavelength-dependent) are required to calculate photolysis frequencies. Pyruvic acid photolysis is not currently calculated in the model because only the photolysis frequencies are available in the literature.

Because the hydroperoxide (-OOH) moiety is expected to be photosensitive, we include photolysis reactions for species bearing this organic function, using the cross-sections and quantum yields measured for $H_2O_2$ (as in Leriche et al., 2003). For further improvement, photolysis reactions will be extended to other compounds when experimental data are available to determine which aqueous phase oxygenated compounds are photosensitive. Epstein et al. (2013) have shown that aqueous photolysis quantum yields are highly dependent on the type of molecule. Using similarity criteria to estimate photolysis rates in the aqueous phase may be too error prone. Furthermore their estimates also show that photolysis would efficiently compete with HO• oxidation for very few of photolabile species. If more data and reliable SAR become available on this subject, a mechanism generated using the present protocol would be the ideal tool to expand on Epstein et al. (2013) study.

## 4. Organic radicals

### 4.1 Alkyl radical O₂ addition

In dilute aqueous solution, alkyl radicals react with dissolved $O_2$ to form peroxyl radicals:

**R.7**    $R \cdot + O_2 \rightarrow R(OO \cdot)$

Recent studies suggest that under high organic radical concentrations, this addition competes with the self- or cross-reactions of alkyl radicals, yielding high molecular weight molecules, such as oligomers (see Ervens et al., 2015; Griffith et al., 2013; Lim et al., 2013; Renard et al., 2015). This competition can favour oligomerization if $O_2$ is not readily available in the aqueous phase. In their bulk aqueous phase modeling study, Ervens et al. (2015) have shown that the aqueous phase under laboratory experiment conditions is not saturated with oxygen, leading to possible oligomerization. Their sensitivity studies however show that oxygen reached saturation in few seconds for atmospheric deliquescent particles, likely because of a large surface to volume ratio. We follow the same hypothesis for cloud droplets.The review by Alfassi (1997) showed that the kinetics of the great majority of $R + O_2$ reactions are close to the diffusion limit, with rate constants of $2.0–4.0 \times 10^9$ $M^{-1}$ $s^{-1}$, and for non-carbon–centred radicals (such as nitrogen-centred radicals), significantly smaller rate constants are observed ($10^7$ to $10^8$ $M^{-1}$ $s^{-1}$). Hence, in our mechanism (which does not consider nitrogen-centred radicals) H-abstraction and $O_2$-addition steps are combined in a single step reaction due to the fast $O_2$-addition reaction rate in the mechanism. In the CAPRAM mechanism (Ervens et al., 2003; Herrmann et al., 2005; Tilgner et al., 2013), the $O_2$-addition step is explicitly introduced. This allows the direct treatment of the alkyl+alkyl vs. alkyl+$O_2$ competition that may occur in deliquesced particles. This is not in the objective of the present work and this consideration will also lead to the implementation in the mechanism of a lot of intermediate chemical species with short lifetime. It could be considered for future versions of the CLEPS mechanism.

### 4.2 Tetroxide formation and decomposition

In general, a peroxyl radical reacts with itself or another peroxyl radical to form a tetroxide, which quickly decomposes (von Sonntag and Schuchmann, 1997). These reactions could be introduced to the mechanism by having each peroxyl radical react with every other peroxyl radical. With 363 peroxyl radicals in the mechanism, this would require more than 66,000 reactions to be written to account for theses cross-reactions. As a first approach, we restrict the mechanism to self-reactions. There are available methods to simplify the description of cross-reactions (Madronich and Calvert, 1990). These methods could be adapted for future versions of the mechanism.

The decomposition of tetroxide follows different pathways, depending on the nature of the initial peroxyl radical. Piesiak et al. (1984) proposed a mechanism for the evolution of the tetroxide formed after dimerization of β-hydroxyethylperoxyl radicals. Zegota et al. (1986) studied the self-reaction of the acetonylperoxyl radical and Schuchmann et al. (1985) explored the fate of the acetate peroxyl radical. The results of these studies have been extended in other experimental works for other peroxyl radicals (Liu et al., 2009; Monod et al., 2007; Poulain et al., 2010; Schaefer et al., 2012; Schöne et al., 2014; Stemmler and von Gunten, 2000; Zhang et al., 2010). We therefore implement peroxyl radical self-reactions following the similarity criteria detailed below.

For β-peroxycarboxylic acids ($>C(OO^{\bullet})C(=O)(OH)$) and their conjugated bases, we generalize the experimental results obtained by Schuchmann et al. (1985) for the acetate peroxyl radical ($CH_2(OO\cdot)C(=O)(O^-)$). For this radical, the tetroxide is degraded through four pathways (reaction R.8):

**R.8** $2\ CH_2(O_2\cdot)CO(O^-)$

$$\xrightarrow{k_{global}=7.5\times10^7\ M^{-1}s^{-1}}\text{tetroxide}\rightarrow\begin{cases}CHOCO(O^-)+CH_2(OH)CO(O^-)+O_2 & \text{branching ratio: 30\%}\\ 2\ CHOCO(O^-)+H_2O_2 & \text{branching ratio: 30\%}\\ 2\ HCHO+2\ CO_2+H_2O_2+2OH^- & \text{branching ratio: 30\%}\\ 2\ CH_2(O\cdot)CO(O^-)+O_2 & \text{branching ratio: 10\%}\end{cases}$$

The four pathways retained in this work are the most important identified by Schuchmann et al. (1985). The sum of these pathways contributes to 87% of the tetroxide decomposition, and each individual contribution is scaled to reach 100% overall.

The evolution of β-hydroxyperoxyl radicals ($>C(OH)C(OO^{\bullet})<$) is represented by the experimental results obtained by Piesiak et al. (1984) for the β-hydroxyethylperoxyl radical ($CH_2(OH)CH_2(OO\cdot)$) (reaction R.9):

**R.9** $2\ CH_2(OH)CH_2(OO\cdot)\xrightarrow{k_{global}=1.0\times10^8\ M^{-1}s^{-1}}\text{tetroxide}\rightarrow\begin{cases}CHOCH_2(OH)+CH_2(OH)CH_2(OH)+O_2 & 33\%\\ 2\ CHOCH_2(OH)+H_2O_2 & 50\%\\ 2\ CH_2(O\cdot)CH_2(OH)+O_2 & 17\%\end{cases}$

The three pathways are reported to contribute to 90% of the degradation of the tetroxide (Piesiak et al., 1984). The mechanism is restricted to these major pathways, and their individual contributions are scaled to reach 100% overall in our mechanism.

β-oxoperoxyl radicals ($-COC(OO^{\bullet})<$) are treated based on the studies of the acetonylperoxyl radical by Poulain et al. (2010) and Zegota et al. (1986) (reaction R.10):

**R.10** $2\ CH_3COCH_2(OO\cdot)\xrightarrow{k_{global}=4.0\times10^8\ M^{-1}s^{-1}}\text{tetroxide}\rightarrow\begin{cases}CH_3COCH_2(OH)+CH_3COCHO+O_2 & 20\%\\ 2\ CH_3COCHO+H_2O_2 & 45\%\\ 2\ CH_3COCH_2(O\cdot)+O_2 & 35\%\end{cases}$

Except for the α-hydroxyperoxyl and acylperoxyl radicals that are discussed in detail in the following subsections, peroxyl radicals that are not included in the above categories are addressed using the estimates from Monod et al. (2007) for the ethylperoxyl radical (reaction R.11):

**R.11**     $2\ CH_3CH_2(OO\cdot) \xrightarrow{k_{global}=1.6\times10^8\ M^{-1}s^{-1}} \text{tetroxide} \rightarrow \begin{cases} 2\ CH_3CHO + H_2O_2 & \quad\quad 20\% \\ 2\ CH_3CH_2O\cdot + O_2 & \quad\quad 80\% \end{cases}$

The rate constant was measured by Herrmann et al. (1999). Schuchmann and von Sonntag (1984) estimated that the first pathway (aldehyde pathway) contributes to 20% of the tetroxide decomposition. Studying the ethylperoxyl radical derived from the photooxidation of ethylhydroperoxyde, Monod et al. (2007) found that the second pathway (alkoxyl pathway) is more likely than the aldehyde pathway, in agreement with previous studies (Henon et al., 1997; von Sonntag and Schuchmann, 1997). Therefore, we attributed the remaining degradation of the tetroxide to the alkoxyl pathway.

### 4.3 α-hydroxyperoxyl HO₂˙/O₂˙⁻ elimination

When a hydroxyl moiety is in the alpha position of the peroxyl function, the peroxyl radical likely undergoes $HO_2^{\bullet}$ elimination:

**R.12**     $>C(OH)(OO\cdot) \rightarrow >C=O + HO_2\cdot$

and an $O_2^{\bullet-}$ elimination following a basic catalysis (Zegota et al., 1986):

**R.13**     $>C(OH)(OO\cdot) + OH^- \rightarrow >C=O + O_2\cdot^- + H_2O$

von Sonntag (1987) showed that the $HO_2^{\bullet}$ elimination rate constant is dependent on the nature of the substituent attached to the carbon atom. Following this study, we generalized the $HO_2^{\bullet}$ elimination rate constants for unknown species using available experimental values. The generalization rules are detailed in **Table 3**.

In the case of α-dihydroxy-peroxyl compounds (-C(OH)(OH)(OO˙)), McElroy and Waygood (1991) showed that $HC(OH)(OH)(O_2^{\bullet})$ decays with a rate constant $k > 10^6$ s$^{-1}$. Without additionnal estimates of $HO_2^{\bullet}$ elimination, we apply the same elimination rate constant of $k = 10^6$ s$^{-1}$ for all α-dihydroxy-peroxyl compounds. Ilan et al. (1976) and Neta et al. (1990) provided an $O_2^{\bullet-}$ elimination reaction rate for the 2,α- hydroxypropylperoxyl radical ($k(CH_3C(OH)(OO^{\bullet})CH_3 + OH^-) = 5.2\times10^9$ M$^{-1}$ s$^{-1}$) and the α-hydroxyethylperoxyl radical ($k(CH_3CH(OH)(OO^{\bullet}) + OH^-) = 4.0\times10^9$ M$^{-1}$ s$^{-1}$). Given the high concentrations of OH$^-$ in water, these reactions are expected to be fast and should not be limiting steps. Therefore, an arbitrary rate constant $k = 4.0\times10^9$ M$^{-1}$ s$^{-1}$, close to the measured rate constants, is assigned to each elimination reaction.

As shown in **Table 3**, $HO_2^{\bullet}$ elimination is a fast process, with an associated lifetime ranging from $1.5\times10^{-3}$ s to 0.1 s. In our simulation (see Section 6) the high range of concentrations for peroxyl radicals is around $10^{-10}$ M for the peroxyl radical derived from glyoxal; tetroxide formation therefore occurs on a timescale of approximately 50 s. Therefore, tetroxide formation and its subsequent decomposition are not considered for α-hydroxyperoxyl radicals, and only direct decomposition is included in the mechanism.

### 4.4 Acylperoxyl decarboxylation

Acylperoxyl radicals (-CO(OO˙)) are treated like standard peroxyl radicals (see R.11) for which only the alkoxyl formation pathway is considered due to the lack of an H-atom on the peroxyl radical. For the acylperoxyl derived from propionaldehyde, this gives:

**R.14**     $2\ CH_3CH_2CO(OO\cdot) \xrightarrow{k=1.6\ 10^8\ M^{-1}s^{-1}} \text{tetroxide} \rightarrow 2\ CH_3CH_2CO(O\cdot) + O_2$

The acylalkoxyl radical undergoes C-C bond scission and yields $CO_2$, accounting for the expected decarboxylation of the acylperoxyl radical.

### 4.5 Alkoxyl radicals

Alkoxyl radicals (RO•) are formed after the decomposition of a tetroxide or after the H-abstraction from a –OH functional group. Their reactivity can proceed through two different pathways, 1-2 hydrogen shift (DeCosta and Pincock, 1989):

**R.15** $>CH(O\cdot)\rightarrow>C\cdot(OH)$

and C-C bond scission (Hilborn and Pincock, 1991):

**R.16** $R\text{-}CH(O\cdot)R'\rightarrow>RCH(=O) + R'\cdot$

Both pathways are non-limiting steps that are in competition with each other. Schuchmann et al. (1985) studied the fate of acetate peroxyl radicals and showed that the produced alkoxyl radical ($CH_2(O^\bullet)C(=O)(O^-)$) may be degraded following R.15 and R.16. However, they could not determine the relative contribution of both reaction pathways to the degradation of the alkoxyl radical. In our mechanism, bond scission (R.16) is the only possible reaction when a neighbouring carbon atom is oxygenated. The scission leads to the formation of the most stable radicals, *i.e.,* the formation of secondary radicals is favoured over the formation of primary radicals. Alkoxyl radicals evolve through a 1-2 hydrogen shift (R.15) when the neighbouring carbon atoms are not oxygenated.

Because of their very short lifetimes, alkoxyl radicals are not explicitly considered in the mechanism. Instead, electron transfer and fragmentation products are directly included in the global reaction. For example, for the β-hydroxyethylperoxyl radical using the reaction rate and branching ratios from Piesiak et al. (1984) (see R.9):

| | | |
|---|---|---|
| Pathway 1: $2\ CH_2(OH)CH_2(OO^\bullet) \rightarrow 2\ CHOCH_2(OH) + H_2O_2$ | $k = 5.0\ 10^7\ M^{-1}\ s^{-1}$ | Piezak et al. (1984) |
| Pathway 2: $2\ CH_2(OH)CH_2(OO^\bullet) \rightarrow CHOCH_2(OH) + CH_2(OH)CH_2(OH) + O_2$ | $k = 3.3\ 10^7\ M^{-1}\ s^{-1}$ | Piezak et al. (1984) |
| Pathway 3: $2\ CH_2(OH)CH_2(OO^\bullet) \rightarrow 2\ CH_2(O^\bullet)CH_2(OH) + O_2$ | $k = 1.7\ 10^7\ M^{-1}\ s^{-1}$ | Piezak et al. (1984) |
| $CH_2(O^\bullet)CH_2(OH) \rightarrow C^\bullet H_2(OH) + CH_2O$ | Fast | |
| $C^\bullet H_2(OH) + O_2 \rightarrow CH_2(OH)(OO^\bullet)$ | $k = 2.0\cdot10^9\ M^{-1}\ s^{-1}$ | |
| $2\ CH_2(OH)CH_2(OO^\bullet) \rightarrow 1.33\ CHOCH_2(OH) + 0.33\ CH_2(OH)CH_2(OH)$ $+ 0.34\ CH_2O + 0.34\ CH_2(OH)(OO^\bullet) + 0.50\ H_2O_2 + 0.16\ O_2 - 0.34\ H_2O$ | $k = 1.0\ 10^8\ M^{-1}\ s^{-1}$ | Piezak et al. (1984) |

The last reaction is the overall budget reaction, which is taken into account in the model.

This treatment of peroxyl and alkoxyl radicals is an attempt at systematizing the approach that is also used in CAPRAM 3.0 (Herrmann et al., 2005; Tilgner et al., 2013). They also consider peroxyl radicals recombination reactions using experimental data from Zegota et al. (1986), Schuchmann et al. (1985) and Poulain et al. (2010). Similarly to our mechanism, the possible cross-reactions are not considered. In the CAPRAM mechanism, alkoxyl radicals can only be fragmented, with a very fast reaction rate following an analogy with gas phase values.

## 5. Coupling CLEPS with MCM v3.3.1 mass transfer

### 5.1 Gas phase mechanism

The CLEPS mechanism is coupled to the gas phase Master Chemical Mechanism, MCM v3.3.1 (Jenkin et al., 2015; Saunders et al., 2003) provided at: http://mcm.leeds.ac.uk/MCM. The new version 3.3.1 of MCM includes in particular the treatment of isoprene oxidation products such as epoxydiols, hydroxymethylmethyl-α-lactone (HMML) and methacrylic acid epoxide (MAE).

All gases are dissolved in CLEPS even if they are not further oxidized in the aqueous phase. Conversely, some aqueous species described in CLEPS can be outgassed even if there is no corresponding gas species in MCM. Among the 87 chemical species included in CLEPS, 33 do not have a counterpart in MCM. These are mostly highly oxygenated and highly soluble species. Conversely, 267 gas phase species from MCM have no corresponding aqueous species in CLEPS. For each species with no equivalent in the other phase, we create an artificial equivalent in the other phase for which no reactivity is

described. The mass transfer parameters are estimated as described below (section 5.2) to accurately determine in which phase the species should reside.

## 5.2 Estimating mass transfer parameters

Mass transfer is described following the kinetic parameterization from Schwartz (1986). For a given species A:

**R.17**
$$\begin{cases} A_{(g)} \xrightarrow{k_I} A_{(aq)} \\ A_{(aq)} \xrightarrow{k_I/H_A} A_{(g)} \end{cases}$$

where $H_A$ [M atm$^{-1}$] is the Henry's law constant for species A and $k_I$ is the pseudo first order rate constant for mass transfer:

**R.18** $\quad k_I = L k_T = L \left( \frac{r^2}{3D_g} + \frac{4r}{3v\alpha} \right)^{-1}$

where $L$ [vol. water/vol. air] is the liquid water content of the cloud, $r$ [cm] is the radius of the droplets, $D_g$ [cm$^2$ s$^{-1}$] is the gas diffusion coefficient, $v$ [cm s$^{-1}$] is the mean molecular speed and $\alpha$ [dimensionless] is the mass accommodation coefficient. The parameters $H_A$, $D_g$, $v$, and $\alpha$ are documented for each soluble species in order to fully describe mass transfer. Estimates of unknown parameters are obtained following the approach of Mouchel-Vallon et al. (2013). The Henry's law coefficients are provided by the GROMHE SAR (Raventos-Duran et al., 2010). Comparing this SAR with other available methods (Meyland and Howard, 2000; Hilal et al., 2008), Raventos-Duran et al. (2010) have shown that GROMHE is the more reliable SAR in general, estimating Henry's law constants with a root mean square error of 0.38 log units (approx. a factor of two). It particularly shows better performances than the other tested methods for the more soluble species, i.e. highly oxygenated, multifunctional organic species.

When unavailable, the temperature dependencies (enthalpy of dissolution) are set to 50 kJ mol$^{-1}$. $D_g$ is calculated by scaling from a reference compound ($\frac{D_{g,A}}{D_{g,H_2O}} = \sqrt{\frac{M_{H_2O}}{M_A}}$; where $D_{g,H_2O} = 0.214/P$ cm$^2$ s$^{-1}$, P [atm] is the atmospheric pressure and M$_A$ is the molar mass [g mol$^{-1}$]; Ivanov et al., 2007). The mean molecular speed is defined as $\sqrt{\frac{8R'T}{\pi M_A}}$ with R'=8.3145 10$^7$ g cm$^2$ s$^{-2}$ K$^{-1}$ mol$^{-1}$. The accommodation coefficients are set to a default value of α = 0.05 when no data are available (Lelieveld and Crutzen, 1991; Davidovits et al., 2006, 2011). We add the temperature dependence of the mass accommodation coefficients based on the parameterization from Nathanson et al. (1996):

**R.19** $\quad \frac{\alpha}{1-\alpha} = e^{-\frac{\Delta G_{obs}}{RT}}$

$\Delta G_{obs} = \Delta H_{obs} - T\Delta S_{obs}$ [J mol$^{-1}$] is interpreted as a the free energy, where $\Delta H_{obs}$ [J mol$^{-1}$] and $\Delta S_{obs}$ [J mol$^{-1}$ K$^{-1}$] are thermodynamic solvation parameters (free enthalpy and entropy) derived by Nathanson et al. (1996) from experimental works on the heterogeneous uptake coefficients performed at different temperatures. When $\Delta H_{obs}$ and $\Delta S_{obs}$ are experimentally available, they are used to estimate the temperature-dependent α, and in other cases, if the value of α is available in the literature, it is used without the temperature dependency.

The mass transcription description in this protocol differs from the coupling between RACM (Stockwell et al., 1997) and CAPRAM 3.0 that is proposed in Herrmann et al. (2005) and Tilgner et al. (2013). Because RACM is a reduced chemical scheme, gas phase species are lumped. Mass transfer therefore occurs between explicit aqueous phase species and fractions of lumped species. A delumping of RACM group compounds is included in the mechanism in the form of equilibrium reactions between the group compound and the standalone species. As an example, the "Ald" model species in RACM

represents all gaseous aldehydes and is considered to be the source of dissolved acetaldehyde, propionaldehyde and butyraldehyde (Herrmann et al., 2005).

The kinetic parameterization in our cloud chemistry model has been used for a long time (Jacob, 1986). The other cloud chemistry models almost always use experimentally measured Henry's law constants. Ervens et al. (2003) proposed to estimate accommodation coefficient based on using a SAR to empirically estimate $\Delta G_{obs}$. As underlined by Ervens et al. (2003), this method should be used very carefully because the data needed to validate this method are very scarce. Future works could focus on (i) the sensitivity of the system to α estimates and (ii) refining the SAR according to the more recent data reported in Davidovits et al. (2011).

### 5.3 Model description

The mechanism resulting from the coupling of CLEPS with MCM v3.3.1 is integrated in a model based on the Dynamically Simple Model for Atmospheric Chemical Complexity (DSMACC; Emmerson and Evans, 2009) using the Kinetic PreProcessor (KPP: see Damian et al., 2002), which has been modified to account for an aqueous phase, as described in the following. The changes are summarized in blue on **Figure 2**.

Aqueous phase reactions are implemented as a new reaction type. Rate constants in units of $M^{-n}\ s^{-1}$ are converted to $molec^{-n}\ cm^{-3n}\ s^{-1}$, depending on the constrained liquid water content. Aqueous phase equilibria are decomposed as forward and backward reactions. This alternative to the total species approach used in other models (Leriche et al., 2000) has the drawback of making the ODE (Ordinary Differential Equations) system stiffer. However, in our simulation, the model handles the stiffness without noticeable numerical issues. Moreover, this approach has the advantage of allowing the explicit treatment of cross-equilibria. The pH therefore evolves dynamically as $H^+$ is explicitly produced and consumed in the equilibrium reactions.

Mass transfer is also implemented as a new reaction type. The rate constants are calculated following Schwartz (1986) and depend on the estimated Henry's law constants, gas diffusion coefficients, mean molecular speeds and accommodation coefficients (see section 5.2).

In DSMACC, TUV 4.5 (Madronich and Flocke, 1997) is used to calculate the photolysis rates in the gas phase (Emmerson and Evans, 2009). The TUV version included in DSMACC was modified to include aqueous phase photolysis reactions (**Figure 2**). To calculate the photolysis coefficients inside the droplets, the clear sky actinic flux values are multiplied by a factor 1.6 (Ruggaber et al., 1997), and the cross-sections and quantum yields are provided from available experimental data (Deguillaume et al., 2004; Long et al., 2013).

Differential equations are solved with a Rosenbrock solver which has been shown to be a reliable numerical method for stiff ODE systems involved in modelling multiphase chemistry (Djouad et al., 2002, 2003).

## 6. Simulation of a test case

### 6.1 Initial conditions

The model is run with the initial and environmental conditions adapted from the low-$NO_x$ situation described by McNeill et al. (2012). Information about the emissions, deposition and initial concentrations of chemical species are provided in **Table 4**. The situation corresponds to summertime conditions, with the simulation starting on the 21st of June. The simulation is located at the sea level and the coordinates used to calculate actinic fluxes are 45.77°N 2.96°E. The main difference with the situation described in McNeill et al. (2012) is that isoprene is the only emitted primary organic compound. To compensate for the decrease in total emitted organic mass, the isoprene emission is increased from $1.5\times10^6$ in McNeill et al. (2012) to $7.5\times10^6$ molec $cm^{-3}\ s^{-1}$ in our work. Furthermore, deposition is added for the major oxidation products of isoprene to prevent

the accumulation of secondary organic species. The temperature is held constant (290K) during the whole simulation. Under these chemical conditions, the gas chemistry simulation is been run for 31 days (see Supplementary Material **SM3**).

At noon on the 31$^{st}$ day of the simulation, a cloud event is simulated with a constant liquid water content of $3\times10^{-7}$ vol. water/vol. air lasting until midnight with a fixed droplet radius of 10 µm. On that day, sunset happens at 6:45 pm (*i.e.* the actinic flux becomes null). This is a permanent cloud simulation and no attempt is made to represent a specific documented cloudy situation. The objective is to test the multiphase mechanism over a long time scale to check that the mechanism is (i) working as intended and (ii) producing chemical effects in both phases. Testing the model over 12h is a first step to evaluate the impacts (or their absence) of detailed organic chemistry on multiphase cloud chemistry. Future studies will use variable environmental conditions that require the consideration of microphysical processes with our multiphase chemical module.

The cloud event is initialized with 1 µM of iron, which is typical concentrations in continental cloud water (Deguillaume et al., 2014), to enable recycling of oxidants by redox cycles involving iron. The initial pH is set to 4 and is free to evolve. The pH quickly reaches 3.2 (see Supplementary Material **SM4**). An additional simulation is performed to consider the aqueous reactivity of dissolved organic species from the gas phase mechanism, and the reactivity of which is not represented in our aqueous phase mechanism. Each of these dissolved organic species reacts with the HO$^{\bullet}$ radicals with a reaction rate of k = $3.8\times10^{8}$ M$^{-1}$ s$^{-1}$. This value is taken from the work of Arakaki et al. (2013), which estimated the sink for aqueous HO$^{\bullet}$ by dissolved organic carbon (DOC). This additional sensitivity test (called "with DOC") is performed to improve the estimate of the HO$^{\bullet}$ concentrations in the atmospheric drops. To account for the conversion of radicals, we assume that each of these reactions produces an HO$_2^{\bullet}$ radical in the aqueous phase.

**6.2 Gas chemical reactivity**

**Figures 3a & 3b** show the time evolution of the targeted gases during the 31$^{st}$ day of the gas phase simulation (dashed lines). The NO$_x$ and O$_3$ mixing ratios (**Figure 3a**) are 0.54 ppbv and 87 ppbv, respectively, at noon while the HO$^{\bullet}$ mixing ratio reaches a maximum of 0.12 pptv. The simulated mixing ratio of isoprene (**Figure 3b**) exhibits a 1.5 ppbv peak in the morning and a minimum of 0.9 ppbv in the afternoon. Because the simulated emission of isoprene is constant during the day (and is turned off at night), its time evolution is constrained by the daytime evolution of its oxidants HO$^{\bullet}$ and O$_3$. In this case, the HO$^{\bullet}$ radical is the main oxidants of isoprene (k$_{HO+isoprene}\times$C$_{HO}$ ≈ $10\times$k$_{O3+isoprene}\times$C$_{O3}$). Therefore, simulated isoprene exhibits a minimum at noon when HO$^{\bullet}$ reaches its maximum. The resulting isoprene diurnal profile is not realistic, as in the atmosphere the isoprene diurnal profile is constrained by the diurnal variation of both its emissions and level of oxidants. The oxidation of isoprene leads to the production of secondary organic species. The time evolutions of the most important secondary species are depicted in **Figure 3b**. The first oxidation products from isoprene (MACR, MVK) follow the same time profile as isoprene. The mixing ratios of other oxidation products vary also temporally depending on their production/destruction rates. For example, MGLY, GLY and glycolaldehyde mixing ratios decrease initially due to their oxidation by HO$^{\bullet}$ and then increase strongly due to their production by the oxidation of isoprene.

**6.3 Impact of aqueous phase reactivity**

**Figures 3a & 3b** show the time evolution of targeted gases during the cloud event (full lines) compared to the gas phase scenario (dashed lines). Previous modelling studies have shown that gas phase HO$_x$ chemistry is modified by the aqueous HO$_2^{\bullet}$ chemistry (Jacob, 1986; Monod and Carlier, 1999). Recent experimental results from Whalley et al. (2015) confirmed that uptake and reactivity in clouds can have a significant impact on the HO$_2^{\bullet}$ and HO$^{\bullet}$ concentrations in the gas phase. In the simulation, at the onset of the cloud event, the HO$_2^{\bullet}$ mixing ratio is reduced by 17%, and the HO$^{\bullet}$ mixing ratios increases by 75%. After an initial sharp decrease, the H$_2$O$_2$ mixing ratio exhibits a 50% increase after 4 hours compared to the dry situation. The increase of the HO$^{\bullet}$ mixing ratios is caused by the important dissolution of organic matter, leading to reduced

HO$^{\bullet}$ sinks in the gas phase. $H_2O_2$ is a soluble species highly reactive with $SO_2$, which explains the initial dip in its mixing ratio. After $SO_2$ is entirely depleted (not shown), the aqueous production of $H_2O_2$ is responsible for its subsequent higher gaseous levels.

This trend in HO$^{\bullet}$ mixing ratios contradicts previous modeling results (Herrmann et al., 2000; Barth et al., 2003; Ervens et al., 2003; Tilgner et al., 2013) which exhibit a decrease in HO$^{\bullet}$ mixing ratios during cloud events. The chosen chemical scenario might be the reason for this difference. Even if the chemical scenario in our study is rather similar to the one in Ervens et al. (2003), we still differ in the amount of emitted organic compounds. In our test simulations, we mainly emit isoprene, with a small contribution of formaldehyde and acetaldehyde, whereas Ervens et al. (2003) emit a larger range of hydrocarbons of anthropogenic (alkanes, alkenes, aromatics) and biogenic origin (limonene, α-pinene). As far as we understand the CAPRAM model setup, these hydrocarbons are not dissolved and it should be noted that they are highly reactive with HO$^{\bullet}$. This means that the large, and certainly major, sink of gaseous HO$^{\bullet}$ caused by hydrocarbons reactivity is always present, even under cloud conditions. When the source of HO$^{\bullet}$ radicals is reduced by the cloud event (*e.g.* due to $HO_2$ and NO separation), HO$^{\bullet}$ radicals sinks are not significantly perturbed and HO$^{\bullet}$ steady state mixing ratios decrease. Conversely, in our simulation the gaseous HO$^{\bullet}$ sink is more significantly perturbed by the cloud event because most of the organic matter in our scenario is produced from isoprene oxidation and is readily soluble. In our case, it seems that the HO$^{\bullet}$ gaseous source reduction is overcompensated by the reduction in HO$^{\bullet}$ gaseous sinks. As a consequence, HO$^{\bullet}$ steady state mixing ratios are higher during cloud events. This hypothesis especially highlights how the chosen chemical scenario and regime is important for simulation results and conclusions. Future work should therefore systematically explore cloud simulations under a large range of scenarios.

Glyoxal, glycolaldehyde, pyruvic acid, glyoxylic acid and glycolic acid are readily soluble species that react in the aqueous phase (Herrmann et al., 2015). Cloud dissolution and oxidation act as significant sinks for these species. For instance, the glyoxal mixing ratios is reduced by 67% at the start of the cloud event, and the glycolaldehyde mixing ratio is significantly reduced until sunset (6:45 PM). For all secondary organic species, daytime gas phase oxidation is increased due to the higher HO$^{\bullet}$ mixing ratios. The addition of aqueous dissolution and aqueous reactive sinks for soluble species explains the sharp decrease in the gas phase mixing ratios for glyoxal, glycolaldehyde, pyruvic acid, glyoxylic acid and glycolic acid. However, the aqueous phase is also a source of secondary organic species. For species that are universal intermediates or end products, the aqueous phase production can be outgassed and contribute to maintaining dry conditions mixing ratios (methylglyoxal, formaldehyde) or significantly increase the mixing ratios compared to dry conditions (acetic and formic acids). Aqueous phase production is also responsible for introducing an infinitesimal amount of oxalic acid ($< 10^{-9}$ ppbv) into the gas phase, as there is no oxalic acid formation pathway in the gas phase. The addition of the missing aqueous oxidation sink for all dissolved species (red lines in **Figures 3a & 3b**) leads to higher concentrations of species for which reactive uptake is an overall sinks (*e.g.*, glyoxal, glycolaldehyde) because the reduced aqueous HO$^{\bullet}$ concentrations (see **Figure 4**) limit the impact of the aqueous sink. In contrast, lower aqueous HO$^{\bullet}$ concentrations reduce the gas phase mixing ratios of species for which the aqueous phase reactivity is an important source (*e.g.*, formic, acetic and glycolic acids).

**Figure 4** shows the time evolution of the main organic aqueous species together with the $H_xO_y$ compounds during the cloud event. The dissolved HO$^{\bullet}$ concentration reaches a peak at $8.5 \times 10^{-14}$ M, which is similar to dissolved the HO$^{\bullet}$ concentrations simulated by Tilgner et al. (2013) for non-permanent clouds in remote conditions and compiled by Arakaki et al. (2013). The oxalic acid concentration is low during the day (approximately $2 \times 10^{-8}$ M) because it is present in the form of iron-oxalate complexes, which are readily photolysed. Therefore, during the night (from 6:45 PM to 12:00 PM), the oxalic acid concentration increases significantly to $10^{-7}$ M.

The sensitivity test including the additional DOC sink shows that the reduced concentration of HO$^{\bullet}$ radicals (from 8.5 to $3 \times 10^{-14}$ M maximum concentration) decreases the sinks of aqueous species from the gas phase (glycolaldehyde,

methylglyoxal and 3,4-dihydroxybutanone), leading to higher aqueous phase concentrations. Conversely, the organic species, which are mostly produced in the aqueous phase (formic, pyruvic, glyoxylic, and oxalic acid) have reduced sources and sinks when HO$^\bullet$ radicals are scavenged by the added DOC. Because sinks and sources of acids due to HO$^\bullet$ radicals should vary in equal proportions, the decrease in organic acids concentrations cannot be ascribed to reactivity with HO$^\bullet$ radicals. We therefore has to consider fixed sinks that do not depend on HO$^\bullet$ concentrations, *i.e.* photolysis and phase transfer. If we consider that acids reach pseudo steady state concentrations, we can assume that because photolysis and phase transfer are not modified by the additional DOC, some acids concentrations could decrease following their overall sources/sinks ratio. MACR and MVK are also less sensitive to the DOC addition. Their main source in water is their mass transfer after gas phase production. This is consistent with their behaviour in the gas phase during the cloud event and could explain why they are less sensitive to the HO$^\bullet$ concentrations.

A detailed budget of aqueous HO$^\bullet$ sinks and sources during the cloud period for the simulation with added DOC (see Supplementary Material **SM5**) shows that $H_2O_2$ is the main source of HO$^\bullet$ *via* the Fenton reaction and its photolysis. However, in the first hours of the cloud event, mass transfer is the major source of HO$^\bullet$, like it was predicted in a previous modeling study on a shorter cloud event considering a remote chemical scenario (Tilgner et al., 2013). Fenton type reactions and photolysis reaction are also significant sources of HO$^\bullet$ in their simulation. The most important overall sink of HO$^\bullet$ radicals in the aqueous phase is the reaction with the added DOC (64%), which results in a slight decrease in the simulated aqueous HO$^\bullet$ concentrations in **Figure 4**. Besides DOC, simulated reactive organics are the most important HO$^\bullet$ sinks, with $C_2$ compounds contributing to 18%, and $C_4$ compounds contributing to 12% of HO$^\bullet$ destruction. $C_1$ and $C_3$ together are responsible for 5% of the HO$^\bullet$ sink. Tilgner et al. (2013) also show that HO$^\bullet$ only aqueous sinks are reactions with organic matter, especially carbonyl compounds such as hydrated formaldehyde, glycolaldehyde and methylglyoxal.

**Figure 5** depicts the contributions in terms of concentrations of the major species in the aqueous phase. The total concentration of organic matter (continuous line) reaches a maximum of 0.76 mM after 12 hours of cloud simulation, which corresponds to approximately 30 mgC L$^{-1}$. This value is high but on the order of magnitude of the DOC measurements (Deguillaume et al., 2014; Giulianelli et al., 2014; Herckes et al., 2013, 2015; van Pinxteren et al., 2015). However, species whose reactivity is represented in the CLEPS aqueous mechanism (dashed line in **Figure 5**) constitute only 16% of the total concentration of dissolved species. Not all species dissolved from the MCM undergo a reactive sink in the aqueous phase (see list in Supplementary Material **SM6**). The 10 most abundant species in the aqueous phase contribute to 91% of the concentration of reactive species (126 *vs.* 138 µM) and 15% of the dissolved species. Among these 10 species, glyoxal, hydroxybutanedione, glycoladehyde, 3,4-dihydroxybutanone and glyoxylic and glycolic acids are the most important contributors. A detailed time-resolved flux analysis of the sources and sinks of these species shows that their initial concentration increase is the result of their mass transfer from the gas phase. Then, balance between aqueous sources and sinks defines the time evolution of their concentrations. For instance, the glyoxal concentration continues increasing because of the important source of the aqueous oxidation of glycolaldehyde. The main sink of glycolaldehyde through reaction with HO$^\bullet$ is strong enough to make its concentration decrease over time. The two most important acids, glycolic and glyoxylic acids, have initial contributions from gas phase mass transfer and are then produced in the aqueous phase from the oxidation of glyoxal and glycolaldehyde. Acetic and formic acids present simulated concentrations that are in the range of *in situ* measurements (Deguillaume et al., 2014). Glycolic and glyoxylic acids present high concentrations in comparison to *in situ* measurements that should indicate that sources or sinks are therefore likely to be misrepresented in our mechanism.

The presence of acids as main contributors to the aqueous phase organic composition shows the potential for cloud reactivity to be a source of acids (Chameides, 1984). The total amount of organic acids (including formic and acetic acids) in both phases is almost doubled in less than an hour by the aqueous phase sources, from approximately 0.48 ppbv of gaseous

organic acids before the cloud event to a total of 0.98 ppbv of organic acids in both phases (see Supplementary Material **SM7**).

**Figure 6** depicts the time evolution of the mean O/C ratio and the mean number of carbon atoms ($n_C$) in the reactive organic compounds present in the aqueous phase (excluding $CO_2$ and iron-organic complexes) and in the gas phase (excluding CO and $CH_4$), with and without the cloud event. The O/C ratios and $n_C$ are a measure of the extent to which long-chain organic species are oxidized and can therefore be a proxy for their functionalization. The reactive aqueous phase is composed of species with an average carbon skeleton of approximately 2.7 carbon atoms and an O/C ratio of 1.1 at the end of the simulation. Large molecules with high functionalization are statistically more soluble than smaller molecules (Mouchel-Vallon et al., 2013; Raventos-Duran et al., 2010). Therefore, at the onset of the cloud event, the larger and more oxygenated species are dissolved, explaining the sharp increase in aqueous $n_C$ at the beginning of the cloud event. In the gas phase, the O/C ratio and $n_C$ follow a marked parabolic curve, reaching a maximum O/C = 0.8 at 15 LT and a minimum $n_C$ = 2.3 at 14 LT. The O/C ratio and $n_C$ then return to the dry-condition levels at sunset. During the day, cloud reactivity is responsible for the increasing O/C ratios and decreasing $n_C$. A higher O/C ratio indicates that the number of oxygenated functions per carbon atom has increased. A lower $n_C$ means that, on average, the organic species have a shorter carbon skeleton as a result of cloud reactivity. This suggests that our aqueous mechanism simulates an efficient fragmentation and functionalization of organic compounds.

At the beginning of the cloud event, many oxygenated and large compounds are dissolved leading to an increase in the O/C ratio and $n_C$ in the gas phase. Then, the reactivity in the aqueous phase generates smaller and more oxygenated species that desorb back to the gas phase, and the increase in the O/C ratio is stronger than under clear sky conditions. The observed effects of aqueous reactivity are confirmed by the addition of DOC, which leads to a slower increase in the O/C ratio and a slower decrease in $n_C$ in both phases because lower $HO^\bullet$ radicals concentrations result in a weaker oxidation capacity of the aqueous phase.

## 7. Conclusions

In this paper we described a new protocol with an explicit chemical scheme for aqueous phase oxidation. This protocol provides an up-to-date method to describe the dissolution of soluble VOCs, their hydration and/or acid dissociation equilibria (as well as iron-oxalate complexation), their reactivity by direct photolysis and their reactivity with $HO^\bullet$ or $NO_3^\bullet$ radicals. It was developed for dilute conditions and low-$NO_x$ conditions and can be generalized to other, more polluted environments by introducing, for example, the multiphase reactivity of organonitrates. In this version, the mechanism includes alkanes, alcohols, carbonyls, carboxylic acids and hydroperoxides. The fate of the newly formed organic radicals is also addressed in detail. The protocol is applied to secondary organic species formed in the aqueous phase. Finally, it is formulated in such a way that it could be implemented in automated chemical scheme generation tools, such as GECKO-A (Aumont et al., 2012; La et al., 2016; Mouchel-Vallon et al., 2013).

Under the simulated cloudy conditions, aqueous phase reactivity is shown to impact the O/C ratio and the size of the secondary organic species, affecting the fragmentation and the functionalization processes resulting from atmospheric oxidation. Furthermore, the addition of a sink for dissolved organic matter shows that this impact on fragmentation and functionalization is sensitive to the aqueous phase oxidative capacity. These simulations were conducted for a long non-realistic permanent cloud. However, the mentioned results are atmospherically relevant, since the impact on O/C ratio and fragmentation can be observed in the first moments of the simulated cloud event.

As long as the mechanism is used to simulate organic chemistry under dilute conditions, such as in cloud droplets, the hypotheses it is built on remains valid. However, modifications should be performed before applying the model to less dilute atmospheric aqueous phases, such as deliquescent aerosols. First, the non-ideality of such aqueous solutions should be taken

into account. Second, H-abstraction and O$_2$-addition should be divided into two distinct steps, and accretion reactions should be considered (Renard et al., 2013). However, the first objective of this work is to thoroughly describe oxidation processes. Accretion processes will be accounted for in future versions of the mechanism. In our simulation, high molecular weight compound precursors, such as glyoxal, glycolaldehyde, and long-chain highly oxygenated species, were dissolved and preferentially formed in the aqueous phase. Models considering radical-initiated oligomer formation are not currently available, except in a few focused cases (Ervens et al., 2015; Woo and Mc Neil 2015). Recent modelling studies implemented newly identified accretion processes to evaluate their potential impacts on SOA formation. However, equivalent accurate knowledge of aqueous oxidation is required since oxidation processes may control the organic radical availability to form accretion products.

Our generated explicit mechanism is based on the Kinetic PreProcessor (KPP) which was built for implementation in chemistry transport 3D models, such as WRF-Chem (Grell et al., 2005) and GEOS-Chem (Eller et al., 2009). The CLEPS mechanism would require some reduction to be coupled with 3D transport models in a similar way to what exists for gas phase chemistry (Emmerson and Evans, 2009; Szopa et al., 2005; Watson et al., 2008) and aqueous phase oxidation (Deguillaume et al., 2009). Several strategies can be used to build reduced multiphase mechanism. Explicit schemes like CLEPS can serve as a reference for the development of reduced chemical schemes like those proposed in Woo and McNeill (2015) or Deguillaume et al. (2009). Very few experimental data are available to validate reduced chemical scheme before implementation in 3D models, especially concerning cloud chemistry. Field measurements (van Pinxteren et al., 2015) and chamber experimental data (Bregonzio-Rozier et al., 2016) are available, but they do not cover the whole range of real atmospheric conditions that a large scale model can encounter. Using explicit modelling tools to serve as reference for a large set of environmental and chemical scenarios is therefore a possible solution to validate reduced multiphase chemical schemes for various environmental conditions. This is of course consistent only once the explicit mechanisms have been validated against the existing experimental data.

This protocol is a powerful tool to explore and propose new reaction mechanisms as a basis to understand experimental studies of scarcely investigated compounds (succinic acid, tartronic acid). The mechanisms generated by our protocol can be used for different purposes in the study of atmospheric aqueous phase processes. They can be evaluated and adapted to laboratory experiments involving a small number of precursors that react only in the aqueous phase. The mechanisms are more likely to be useful for larger-scale experiments involving longer timescales and two phases (gas + water) in environmental cloud chambers (Brégonzio-Rozier et al., 2016). They are also of interest for the modelling studies of field campaigns such as HCCT (Whalley et al., 2015) or SOAS (Nguyen et al., 2014). The SOA and the cloud chemistry communities are currently interested in studying the respective contributions of oxidation and accretion processes to the transformations of organic matter in the aqueous phase and to the oxidative capacity of clouds. Most recent modelling studies have focused on implementing newly identified accretion processes to evaluate their potential impacts on SOA formation (Ervens et al., 2015; McNeill, 2015; McNeill et al., 2012; Woo and McNeill, 2015). In this work, guidelines are developed to update oxidation mechanisms that will be compared in the future to descriptions of the formation of accretion products, such as oligomers, organonitrates and organosulphates.

**Code availability**

The mechanism used in this paper is available in KPP format upon request to l.deguillaume@opgc.univ-bpclermont.fr. Any suggestions and corrections to the mechanism (*e.g.*, new experimental rate constant we may have missed, typos) are also welcomed at the same address. The modified version of DSMACC (originally downloaded at https://github.com/barronh/DSMACC) that was used for the simulations is also available upon request to l.deguillaume@opgc.univ-bpclermont.fr.

## Acknowledgments

The authors acknowledge the French National Agency for Research (ANR) project CUMULUS ANR-2010-BLAN-617-01 for providing financial support. The authors are very grateful to the Agence Nationale de la Recherche (ANR) for its financial support through the BIOCAP project (ANR-13-BS06-0004). Part of this work was also supported by CEA/CNRS through the contract CEA 12-27-C-DSPG/CAJ – CNRS 77265.

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

**Table 1: Experimental hydration constants for carboxylate species.**

| Species | $K_{hyd}$ | References |
|---|---|---|
| **Glyoxylate** | | |
| $CH(=O)C(=O)(O^-)$ | 67 | Tur'yan (1998) |
| **Monoethyl Oxaloacetate** | | |
| $CH_3CH_2\text{-}O\text{-}C(=O)C(=O)CH_2C(=O)(O^-)$ | 3.125 | Kozlowski and Zuman (1987) |
| **Pyruvate** | | |
| $CH_3C(=O)C(=O)(O^-)$ | 0.0572 | Pocker et al. (1969) |
| **α-ketobutyrate** | | |
| $CH_3CH_2C(=O)C(=O)(O^-)$ | 0.08 | Cooper and Redfield (1975) |
| **α-ketoisovalerate** | | |
| $CH_3C(CH_3)C(=O)C(=O)(O^-)$ | 0.075 | Cooper and Redfield (1975) |
| **β-fluoropyruvate** | | |
| $CH_2(F)C(=O)C(=O)(O^-)$ | 19 | Hurley et al. (1979) |

**Table 2: Examples of the reduction scheme applied to estimate HO• reactions branching ratios. $k_i$ values are the partial reaction rate corresponding to the labeled atoms $i$ in the left column.**

| Molecule | Estimated H-abstraction rates $k_i$ on atom labeled **i** following Doussin and Monod (2013) [M$^{-1}$ s$^{-1}$] | Contribution to the global reaction rate | Reactivity threshold considered | Retained pathways | Scaled Contribution | Final estimated rate constants [M$^{-1}$ s$^{-1}$] |
|---|---|---|---|---|---|---|
| **Hydrated Glycolaldehyde**  | $k_1 = 2.9\times10^8$ <br> $k_2 = 0.9\times10^8$ <br> $k_3 = 3.4\times10^8$ <br> $k_4 = 4.0\times10^8$ <br><br> $k_{Global} = 1.1\times10^9$ | 26 % <br> 8 % <br> 30 % <br> 36 % | C$_2$ : 90 % | Yes <br> No <br> Yes <br> Yes | 28 % <br> - <br> 33 % <br> 39 % | $k_1 = 3.1\times10^8$ <br> - <br> $k_3 = 3.6\times10^8$ <br> $k_4 = 4.3\times10^8$ <br><br> $k_{Global} = 1.1\times10^9$ |
| **3-hydroxypropionaldehyde**  | $k_1 = 1.5\times10^9$ <br> $k_2 = 9\times10^7$ <br> $k_3 = 7\times10^7$ <br> $k_4 = 2.5\times10^9$ <br><br> $k_{Global} = 4.1\times10^9$ | 36 % <br> 2 % <br> 2 % <br> 60 % | C$_3$ : 75 % | Yes <br> No <br> No <br> Yes | 38 % <br> - <br> - <br> 62 % | $k_1 = 1.5\times10^9$ <br> - <br> - <br> $k_4 = 2.6\times10^9$ <br><br> $k_{Global} = 4.1\times10^9$ |
| **2-hydroxy, 3-oxobutanoate**  | $k_1 = 8.1\times10^7$ <br> $k_2 = 6.1\times10^7$ <br> $k_3 = 8.5\times10^7$ <br> $k_4 = 9.2\times10^7$ (*) <br><br> $k_{Global} = 3.2\times10^8$ | 25 % <br> 19 % <br> 27 % <br> 29 % | C$_4$ : 75 % | Yes <br> No <br> Yes <br> Yes | 31 % <br> - <br> 33 % <br> 36 % | $k_4 = 1.0\times10^8$ <br> - <br> $k_3 = 1.1\times10^8$ <br> $k_4 = 1.1\times10^8$ <br><br> $k_{Global} = 3.2\times10^8$ |

(*) electron transfer reaction

**Table 3: Generalization of HO$_2^{\bullet}$ elimination rate constants for unknown species, following von Sonntag (1987).**

| | von Sonntag (1987) compilation | | |
|---|---|---|---|
| 1$^{st}$ Substituent | 2$^{nd}$ Substituent | HO$_2^{\bullet}$ elimination rate constant k (s$^{-1}$) | Generalization |
| H | H | <10 | - |
| H | CH$_3$ | 52 | Primary peroxyl radicals |
| H | CH$_2$(OH) | 190 | β-hydroxyperoxyl radicals |
| CH$_3$ | CH$_3$ | 665 | Secondary peroxyl radicals |

**Table 4: a) Chemical scenario used for the gas phase simulation of 31 days. b) Aqueous phase initial concentration.**

a)

| Gas phase species | Initial mixing ratio [ppb] | Emission [molec cm$^{-3}$ s$^{-1}$] | Deposition [s$^{-1}$] |
|---|---|---|---|
| SO$_2$ | 1 | $2.91\times10^5$ | $1\times10^{-5}$ |
| NO | - | $2.86\times10^5$ | - |
| NO$_2$ | 0.3 | - | $4\times10^{-6}$ |
| N$_2$O$_5$ | - | - | $2\times10^{-5}$ |
| HNO$_3$ | 0.3 | - | $2\times10^{-5}$ |
| O$_3$ | 40 | - | $4\times10^{-6}$ |
| H$_2$O$_2$ | 1 | - | $1\times10^{-4}$ |
| CH$_4$ | $1.7\times10^3$ | - | - |
| CO$_2$ | $3.57\times10^5$ | - | - |
| CO | $1.5\times10^2$ | $3.7\times10^6$ | $1\times10^{-6}$ |
| Isoprene | 1 | $7.50\times10^{6}$ [a] | - |
| Dihydroxybutanone | - | - | $5\times10^{-5}$ |
| MACR | - | - | $5\times10^{-5}$ |
| MVK | - | - | $5\times10^{-5}$ |
| Glyoxal | 0.1 | - | $5\times10^{-5}$ |
| Methylglyoxal | 0.1 | - | $5\times10^{-5}$ |
| Glycolaldehyde | - | - | $5\times10^{-5}$ |
| Acetaldehyde | 0.1 | $3.17\times10^3$ | $5\times10^{-5}$ |
| Formaldehyde | 0.5 | $3.03\times10^3$ | $5\times10^{-5}$ |
| Acetone | 0.1 | $8.92\times10^3$ | $5\times10^{-5}$ |
| Pyruvic Acid | - | - | $5\times10^{-5}$ |
| Acetic Acid | $1\times10^{-3}$ | $3.35\times10^3$ | $5\times10^{-5}$ |
| Formic Acid | - | - | $5\times10^{-5}$ |
| Methanol | 2 | $1.07\times10^4$ | $5\times10^{-5}$ |
| Methylhydroperoxide | 0.01 | $3.35\times10^3$ | $5\times10^{-6}$ |

[a] = 0 at nighttime concentration,

b)

| Aqueous phase species | Initial concentrations [μM] |
|---|---|
| Fe$^{2+}$ | 1 |

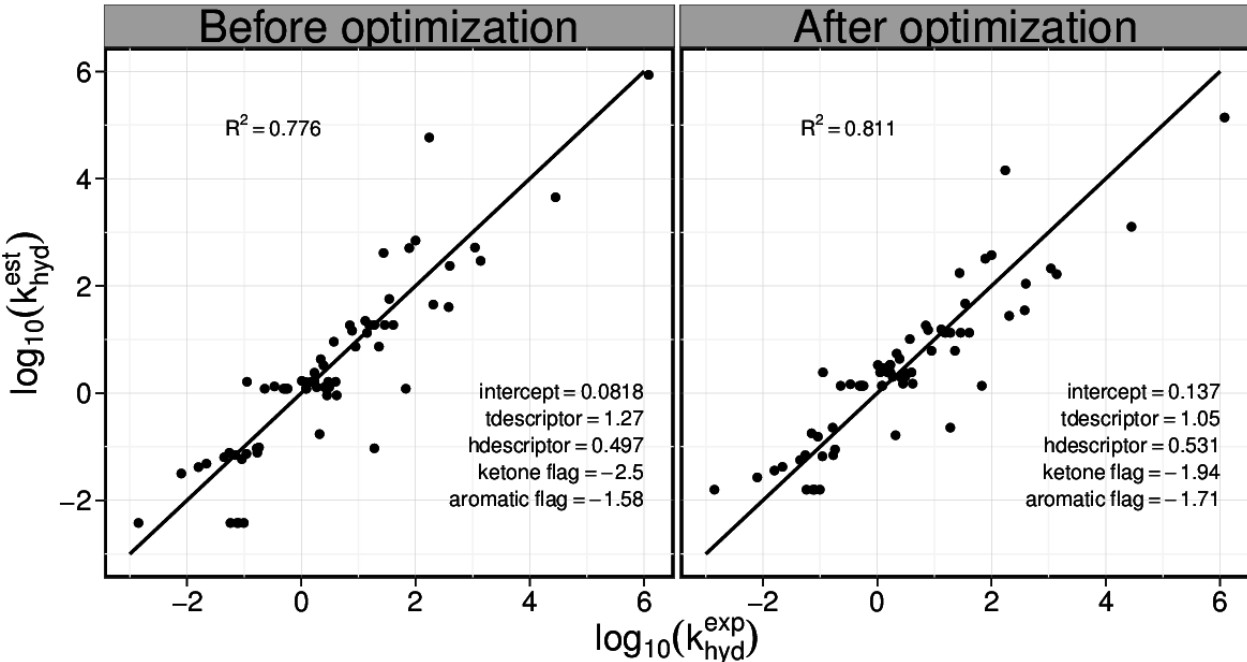

**Figure 1: Scatterplots of the estimated log(K$_{hyd}$) using the SAR from Raventos-Duran et al. (2010) versus the experimental log(K$_{hyd}$), before (left panel) and after (right panel) the optimization for carboxylates. Values for the optimized descriptors are shown on the bottom right of each panel. The values chosen before the optimization are taken from Raventos-Duran et al. (2010). The line is the $y = x$ line.**

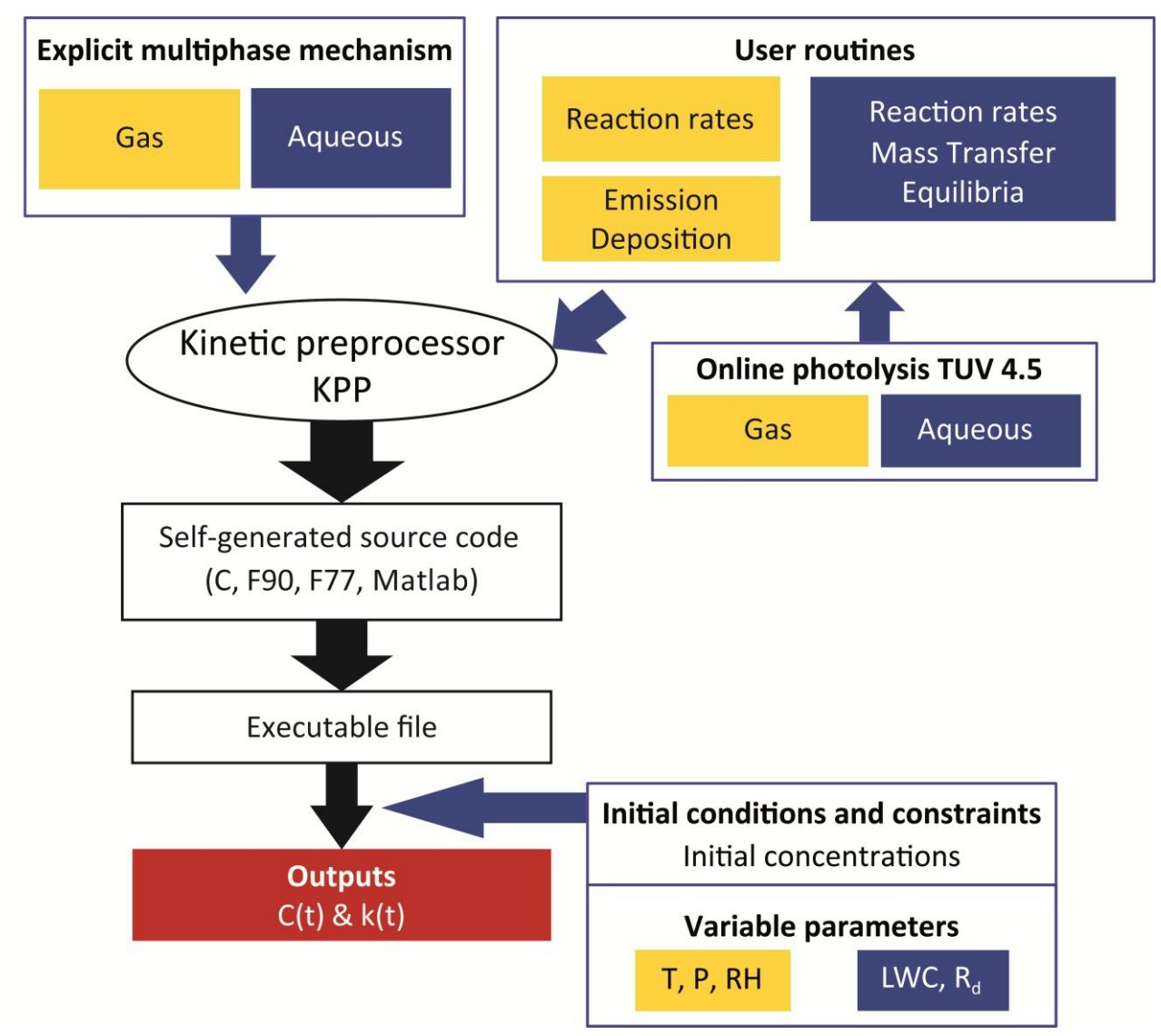

**Figure 2: Schematic diagram of the DSMACC version of the Kinetic PreProcessor. The developments related to aqueous-phase reactivity are shown in blue.**

a)

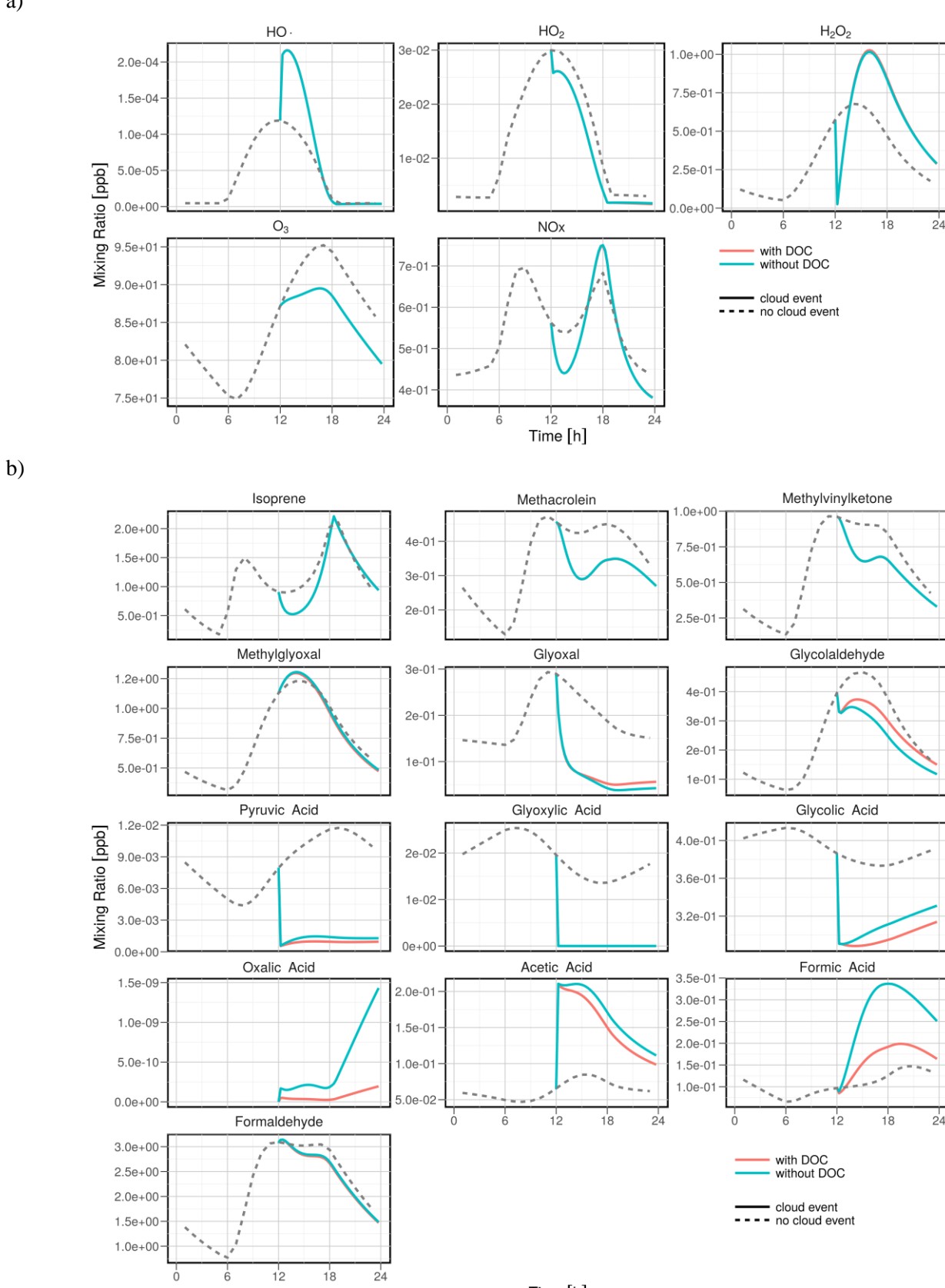

b)

5    **Figures 3a & b: Time evolution of the gas phase mixing ratios without the cloud event (dashed lines) and during the cloud event (continuous line). The cloud event simulations are depicted with (red lines) and without (blue lines) DOC. Please note that for most plots, the red line is hidden by the blue line.**

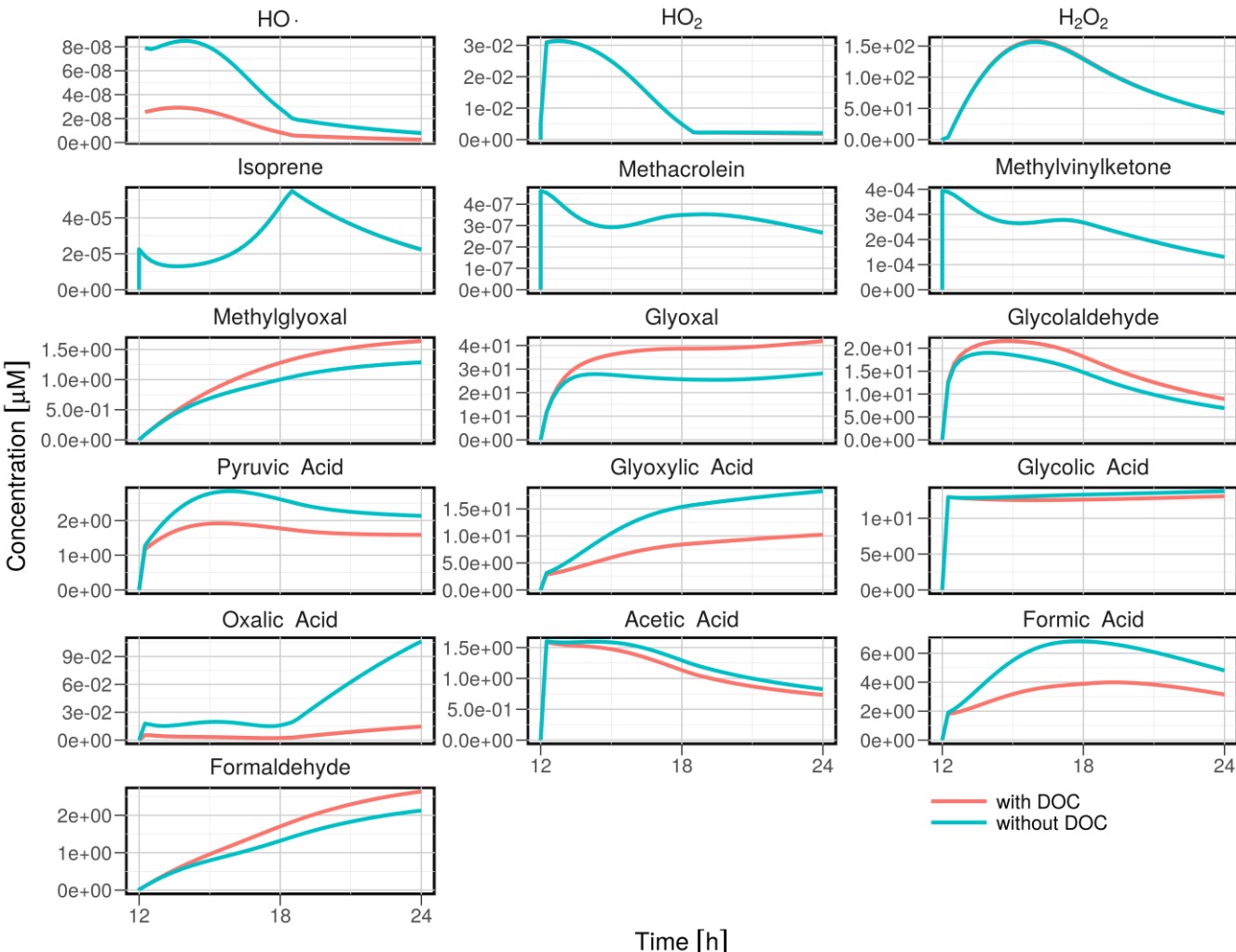

**Figure 4: Time evolution of the dissolved species concentrations during the simulated cloud event with (red lines) and without (blue lines) DOC. The vertical scale is in µM; therefore, the HO• radical concentrations are in the $10^{-14}$ M range. Please note that for some plots, the red line is hidden by the blue line.**

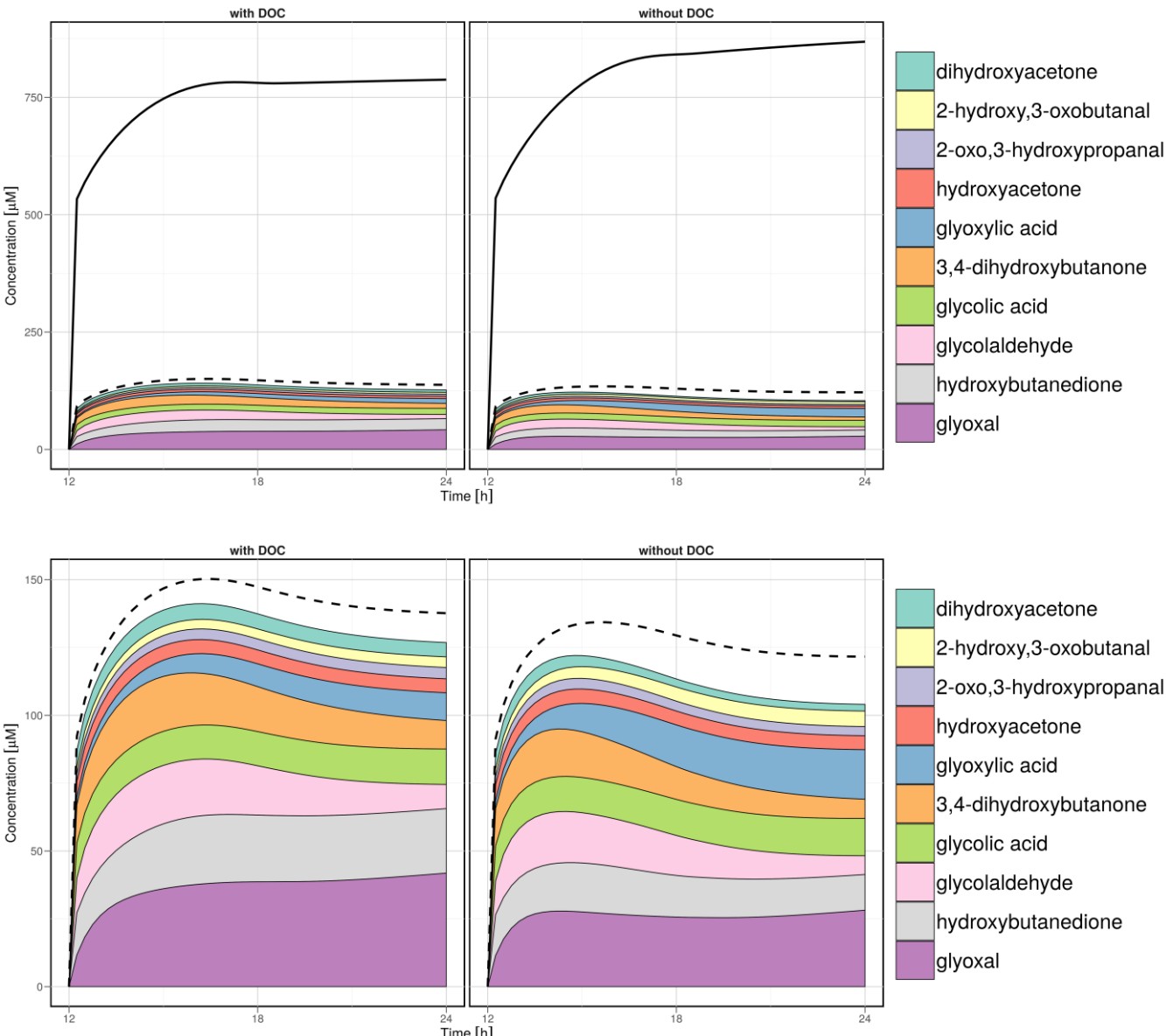

**Figure 5: Contribution of the 10 most important species (in terms of concentrations) in the aqueous phase (colours). The solid line depicts the total concentration of dissolved organic compounds. The dashed line depicts the total concentration of reactive dissolved species.**

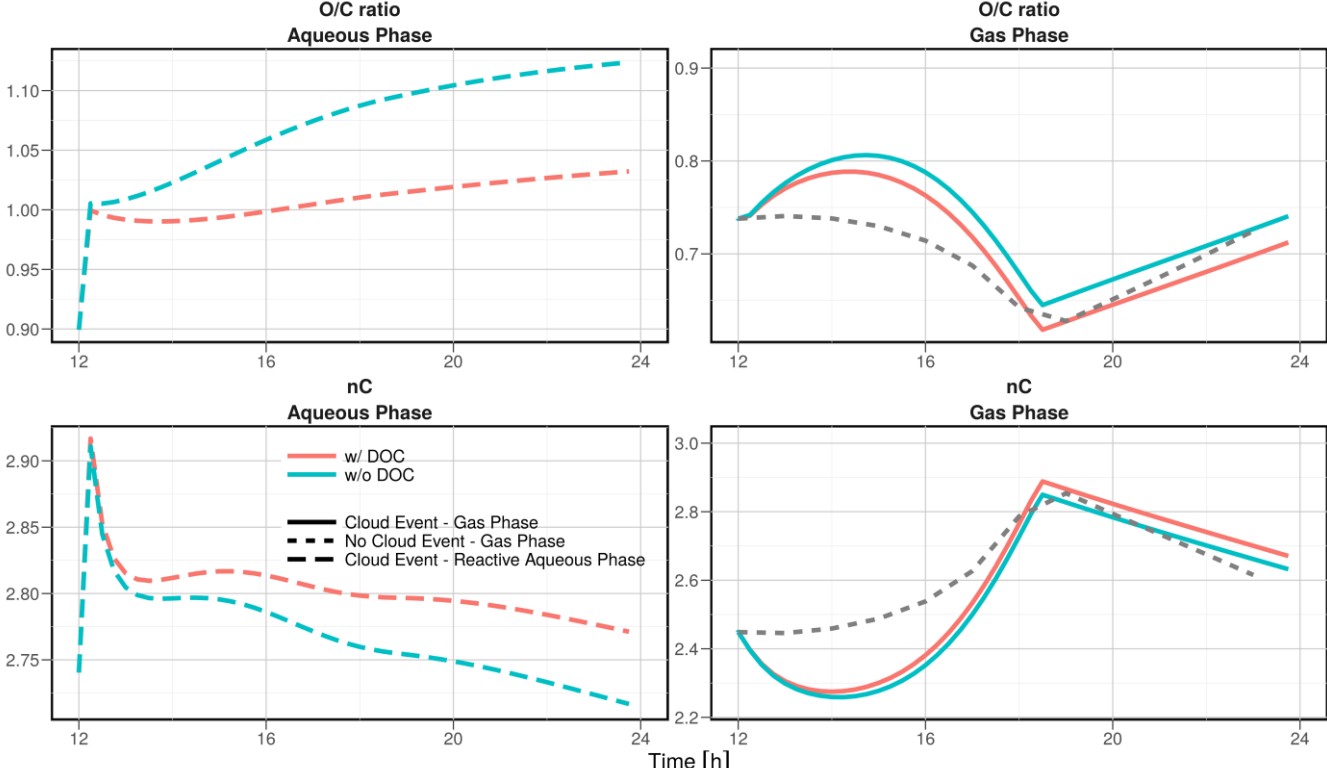

**Figure 6: Time evolution of the mean O/C ratio for the reactive species (top; ratios are calculated on a number of atoms basis; CO, CO₂, CH₄, and iron-organic complexes are excluded from the calculation) and the mean number of carbon atoms $n_C$ (bottom) in the aqueous phase (left) and in the gas phase (right) for the gas phase simulation (short dashed lines) and the cloud event simulation (continuous lines). Cloud event simulations are depicted with (red lines) and without (blue lines) DOC.**