# Peer review of "CLEPS 1.0: A new protocol for cloud aqueous phase oxidation of VOC mechanisms"

_Geoscientific Model Development, 2016_

## Referee Comment (RC1) · Anonymous Referee #1 · 21 Oct 2016

The authors present a 'Cloud Explicit Physicochemical Scheme' that describes the chemical processing of organic and inorganic compounds in the atmospheric aqueous phase of cloud droplets. This is a timely topic since for decades, atmospheric gas phase chemistry has been described in detail whereas chemical reactions in the aqueous phase have not gained as much attention due to the lack of data and its complexity that hampers a comprehensive inclusion in models. Special emphasis is given here to the treatment of organic compounds and a method is presented to estimate branching ratios of C2-C4 compounds and to track all their oxidation products. For about 15 years, another aqueous phase mechanism exist, CAPRAM (in various versions) that can be considered the 'standard' in atmospheric multiphase modeling. While the authors cite some of the papers of CAPRAM development and application, the fail to highlight the similarities of the mechanisms (it seems to me that the inorganic chemistry has been completely taken from CAPRAM) and to discuss the consequences of the additions/changes. The modifications of the aqueous phase mechanism here are based on several assumptions – but so are also some of the parameters in CAPRAM. A discussion is missing of the extent to which the new assumptions are more reliable and necessary and therefore lead to a more consistent aqueous phase mechanism. In summary, in my opinion the manuscript lacks originality and novelty and fails to discuss uncertainties in the current mechanism and similarities/differences to previous work. I might recommend publication of this paper if the following comments are thoroughly addressed.

Main comments

1) Motivation of the current study

The intention of developing a 'complete' and 'correct' aqueous phase mechanism is clearly an ambitious and laudable project. However, it is clear that due to the lack of data many assumptions have to be made that should be better justified here. While such a mechanism can be used in a box model to assess chemical interactions in the multiphase system, it is impossible to track hundreds of species in a larger scale model. Therefore, I am missing a clearer direction on how this goal can be reached. For example, previous studies have suggested several strategies to reduce chemical mechanisms (Ervens et al., 2003; Ervens et al., 2008; Deguillaume et al., 2009; Woo and McNeill, 2015) for use in larger models. One way of reducing models is identifying the most important reactions and species. Such a discussion is missing here. – Instead as further directions it is suggested to include more species (succinic acid, tartronic acid).

2) Similarities to CAPRAM

a) In the abstract and also later, it is claimed that 'a new detailed aqueous phase mechanism . . . is proposed'. Given that all inorganic chemistry seems the same as other mechanisms (such as CAPRAM) and also all 'overall' rate constants are the

same, I think this statement is highly exaggerated. I suggest marking in the reaction tables those processes and parameters that are new or different.

b) It seems to me that several simplifications that have been tested in CAPRAM (Ervens et al., 2003) are repeated here – without the previous detailed sensitivity studies. Such simplifications include skipping the formation of peroxy radicals as an individual step (p. 9, l. 1-5), the self-recombination of peroxy radicals (p. 9, l. 10) and skipping of the tetroxide (p. 9-10). The previous work should be properly referenced here.

c) How does the analysis of OH(aq) sinks and sources (p. 15, top) compare to previous studies such as CAPRAM (Herrmann et al., 2000; Ervens et al., 2003; Deguillaume et al., 2009; Tilgner et al., 2013)?

d) How does the modeled evolution of the O/C ratio compare to the trends as shown by (Schrödner et al., 2014)? Can the differences be ascribed to the more complex formulation of the branching ratios of the organic reactions?

3) Basis of assumptions

a) Assumptions on equilibria (hydration, dissociation, gas/aqueous partitioning)

In Sections 3.1, 3.2 and 5.2, it is explained how unknown equilibrium constants are estimated. While it is clearly necessary to estimate such parameters due to the lack of their availability, it should be shown (i) the validity of these assumptions, (ii) a comparison of the estimated parameters to (the few) known parameters and (iii) which assumptions have been made in other aqueous phase mechanisms and whether these assumptions are less appropriate.

b) I understand that a completely consistent coupling of MCM and the aqueous phase mechanism is difficult. However, the fact that numerous species do not have a gas or aqueous phase equivalent, respectively (p. 11, Section 5.1) introduces an unnecessary inconsistency. In my opinion, it would be more consistent to introduce a solubility threshold (Henry's law constant) that determines where a species should reside.

c) How well do calculated Henry's law constants based on GROMHE SAR compare to measured values? Can you refer to some comparison in a previous study or show it here (possibly in supplemental information)?

4) Inconsistencies in the presented mechanism

a) Dissociation equilibria

- In Eq. 4, the resulting KA constant should be dimensionless. In the Equilibria table in the supplemental information, data are given in M (according to literature). The unit is missing in this table and therefore it is very confusing and leads to a bias of a factor 18 (molecular weight of water) in all dissociation constants. How have the numbers in the table been used in the model?

- Later in the manuscript (p. 12, l. 28), it is stated that the equilibria are split into forward and back reaction. What values are used? How were they derived?

b) Table 2 I do not understand the third example in Table 2. The 'global rate constant' is given as 3.2e8 M-1 s-1; however, all rate constants for the various branching pathways exceed this number and yield a value that is an order of magnitude greater. Is this just a typo or were these (and possibly other?) rate constants used like this used in the model?

5) Discussion of results

a) Length of cloud In several previous model studies, it has been discussed in detail that cloud processing time is on a time scale on the order of a few minutes or at most an hour as it is restricted by the lifetime of a droplet (e.g., Feingold and Kreidenweis, 2002; Ervens et al., 2004; Tilgner et al., 2013). Therefore, the conclusions of the aqueous phase on the oxidation capacity of the atmosphere should be revised and related to more atmospherically relevant conditions.

b) OH profiles Several previous studies have shown that the OH concentration in the gas phase is substantially reduced in the presence of clouds (Herrmann et al., 2000;

Ervens et al., 2003; Tilgner et al., 2013). The current mechanism shows a doubling of OH(gas) during cloud. Admittedly, the initial conditions were slightly different than in the CAPRAM simulations. However, given the robustness of the previous results for various scenarios, the differences of the current study to previous ones should be discussed.

c) How do the differences between the three scenarios in Fig. 3b compare to previous model studies? Can possible differences be ascribed to the new organic pathways?

6) Previous literature on aqueous phase chemistry

- p. 2, l. 12: In previous literature (Herrmann et al., 2015, and references therein) it is discussed that the presence of an aqueous phase (cloud) leads to the separation of the soluble HO2 and the rather insoluble NO in the gas phase. Since their reaction in suppressed, less OH is formed. Therefore, it is not radical chemistry in the aqueous phase but different pathway strengths in the gas phase that lead to differences in OH during cloud.

- p. 2, l. 3: The statement that photolysis rates are 'highly enhanced in clouds' is not always true. While in thin clouds, photolysis rates are enhanced, in dense cloud they might be reduced (cf e.g. Fig. 2 in Ervens, 2015).

- p. 3, l. 5: In the cited paper (Ervens et al., 2015) it is discussed that oligomerization of MACR and MVK is rather unimportant in the atmosphere because of the low solubility of these precursors.

- p. 6, l. 30ff: How well does the product distribution estimated here matches the detailed laboratory study (Perri et al., 2009)?

- p. 8, Section 3.6: Results of a recent key paper on aqueous photolysis should be discussed here (Epstein et al., 2013).

- p. 8, l. 34-36: In the study by Ervens et al., 2015 it is stated "Unlike in laboratory experiments, atmospheric aqueous aerosol particles can be considered saturated with

oxygen (∼270 $\mu$M) due to their large surface–volume ratio. In all our model sensitivity studies with the multiphase model, the oxygen concentration reached saturation level after a few seconds." – which is not clearly reflected in the text in the current manuscript.

- p. 14, l. 37ff: A lower OH concentration will not only lead to lower formation rates of these acids but also to lower destruction rates (Ervens et al., 2014). Therefore the role of OH for the total organic acid levels depends on the ratio of k(formation) /k(destruction), which, in turn, might be a function of pH. This should be mentioned here.

Technical and minor comments

p. 5, l. 21: 'Rate constants'

p. 7, l. 24; p. 8, l. 14, and some other places in the manuscript: 'Reaction rates' are defined as the product of a rate constant and reactant concentration(s). It should be 'rate constant'.

p. 10, l. 28: Give a reference for this concentration.

p. 15, l. 26: Which acids are included in 'total acids'? Do they include formic and acetic acid?

Table 1: The references below the table should be added to the main reference list and not listed here.

Table 4: What parameter does the footnote (b) refer to?

Figure 2, caption: 'Relative' should be replace by 'related'

Fig. 3 and 4: Add in the caption that the red line is hidden by the blue one or even choose different line types (e.g., dotted vs solid)

Fig. 5: Clarify the caption: '10 most important species' in terms of what?

[Figure]

References

Deguillaume, L., Tilgner, A., Schrödner, R., Wolke, R., Chaumerliac, N., and Herrmann, H.: Towards an operational aqueous phase chemistry mechanism for regional chemistry-transport models: CAPRAM-RED and its application to the COSMO-MUSCAT model, J. Atmos. Chem., 64, 1,1-35, 10.1007/s10874-010-9168-8, 2009.

Epstein, S. A., Tapavicza, E., Furche, F., and Nizkorodov, S. A.: Direct photolysis of carbonyl compounds dissolved in cloud and fog droplets, Atmos. Chem. Phys. Discuss., 13, 4,10905-10937, 10.5194/acpd-13-10905-2013, 2013.

Ervens, B., Carlton, A. G., Turpin, B. J., Altieri, K. E., Kreidenweis, S. M., and Feingold, G.: Secondary organic aerosol yields from cloud-processing of isoprene oxidation products, Geophys. Res. Lett., 35, 2,L02816, 10.1029/2007gl031828, 2008.

Ervens, B., Feingold, G., Frost, G. J., and Kreidenweis, S. M.: A modeling study of aqueous production of dicarboxylic acids, 1. Chemical pathways and speciated organic mass production, J. Geophys. Res. - Atmos., 109, D15,D15205, doi: 10.1029/2003JD004387, 2004.

Ervens, B., George, C., Williams, J. E., Buxton, G. V., Salmon, G. A., Bydder, M., Wilkinson, F., Dentener, F., Mirabel, P., Wolke, R., and Herrmann, H.: CAPRAM2.4 (MODAC mechanism): An extended and condensed tropospheric aqueous phase mechanism and its application, J. Geophys. Res., 108, D14,4426, doi: 10.1029/2002JD002202, 2003.

Ervens, B., Renard, P., Tlili, S., Ravier, S., Clément, J. L., and Monod, A.: Aqueous-phase oligomerization of methyl vinyl ketone through photooxidation – Part 2: Development of the chemical mechanism and atmospheric implications, Atmos. Chem. Phys., 15, 16,9109-9127, 10.5194/acp-15-9109-2015, 2015.

Ervens, B., Sorooshian, A., Lim, Y. B., and Turpin, B. J.: Key parameters controlling OH-initiated formation of secondary organic aerosol in the aqueous phase (aqSOA), J.

Geophys. Res. - Atmos., 119, 7,3997-4016, 10.1002/2013JD021021, 2014.

Feingold, G., and Kreidenweis, S.: Cloud Processing of Aerosol as Simulated by a Large Eddy Simulation with Coupled Microphysics and Aqueous Chemistry, J. Geophys. Res., 107, D23,doi: 10.1029/2002JD002054, 2002.

Herrmann, H., Ervens, B., Jacobi, H.-W., Wolke, R., Nowacki, P., and Zellner, R.: CAPRAM2.3: A Chemical Aqueous Phase Radical Mechanism for Tropospheric Chemistry, J. Atmos. Chem. , 36,231-284, 2000.

Herrmann, H., Schaefer, T., Tilgner, A., Styler, S. A., Weller, C., Teich, M., and Otto, T.: Tropospheric Aqueous-Phase Chemistry: Kinetics, Mechanisms, and Its Coupling to a Changing Gas Phase, Chemical Reviews, 115, 10,4259-4334, 10.1021/cr500447k, 2015.

Perri, M. J., Seitzinger, S., and Turpin, B. J.: Secondary organic aerosol production from aqueous photooxidation of glycolaldehyde: Laboratory experiments, Atmos. Environ., 43,1487-1497, 2009.

Schrödner, R., Tilgner, A., Wolke, R., and Herrmann, H.: Modeling the multiphase processing of an urban and a rural air mass with COSMO–MUSCAT, Urban Climate, 10, Part 4, 0,720-731, http://dx.doi.org/10.1016/j.uclim.2014.02.001, 2014.

Tilgner, A., Bräuer, P., Wolke, R., and Herrmann, H.: Modelling multiphase chemistry in deliquescent aerosols and clouds using CAPRAM3.0i, J. Atmos. Chem., 70, 3,221-256, 10.1007/s10874-013-9267-4, 2013.

Woo, J. L., and McNeill, V. F.: simpleGAMMA v1.0 – a reduced model of secondary organic aerosol formation in the aqueous aerosol phase (aaSOA), Geosci. Model Dev., 8, 6,1821-1829, 10.5194/gmd-8-1821-2015, 2015.

---

## Short Comment (SC1) · 24 Oct 2016

Dear authors,

In my role as Executive editor of GMD, I would like to bring to your attention our Editorial version 1.1:

http://www.geosci-model-dev.net/8/3487/2015/gmd-8-3487-2015.html

This highlights some requirements of papers published in GMD, which is also available on the GMD website in the 'Manuscript Types' section:

http://www.geoscientific-model-development.net/submission/manuscript_types.html

In particular, please note that for your paper, the following requirement has not been met in the Discussions paper:

[Figure]

- "The main paper must give the model name and version number (or other unique identifier) in the title."

Please add a version number for CLEPS in the title upon your revised submission to GMD.

Yours,

Astrid Kerkweg
* * *

---

## Referee Comment (RC2) · Anonymous Referee #2 · 22 Nov 2016

Summary
This paper introduces a new protocol for developing a detailed aqueous chemistry mechanism. It emphasizes oxidation by OH, NO$_3$, and other oxidants in low NO$_x$ and dilute conditions, and does not include accretion processes. The detailed aqueous chemistry mechanism, coupled with the Master Chemical Mechanism (MCM v.3.3.1), is applied to an ideal cloud situation to examine the behavior of the chemistry.

This is an exciting, new advancement for our understanding of aqueous chemistry, especially for the oxidation of organic compounds dissolved in cloud water. The paper is fairly well written, although several spots need to be clarified. The paper can be improved by providing more information on the test case, giving more discussion on how the mechanism and results compare to previous studies (with and without detailed aqueous chemistry), and cleaning up the presentation of the paper. My suggestions of needed clarifications are given below.

Specific Comments
1. Page 6, line 25 and Table 2. It is not completely clear to me what criteria are used to proceed with the reduction scheme. For example, why is a branch with a 19% contribution to the reaction removed? Is it simply because the other 3 pathways represent >75% of the global reaction rate? It seems that pathways that comprise 10% or less of the reaction rate can be ignored without much impact on the overall reaction scheme. What kind of impact occurs when the pathway contribution is larger, such as the 19% for the third example in Table 2?
2. Section 5. How does the CLEPS mechanism compare to previous aqueous chemistry studies (e.g. CAPRAM)?
3. Page 13, line 10. While Rosenbrock solvers have become commonplace in chemistry transport models, other solver techniques also adequately solve the gas-aqueous chemistry mechanism (e.g., Ervens et al., 2003, JGR; Barth et al., 2003, JGR; McNeill et al., 2012, ES&T).
4. Page 13, line 12. Please restate the objective of section 6. The introduction says that "the box model is tested for an ideal cloud situation", which implies some kind of evaluation. However, there is no evaluation of the model results provided and there is only one instance of a comparison with previous modeling work (page 14, line 30).
5. As a follow up to the previous comment, please include more discussion in Section 6 of how the model results in this paper compare to previous ones, e.g. McNeill et al., 2012; Tilgner et al., 2013; Herrmann et al., 2005 – which can hopefully be directly compared due to their similarity of the test case conditions, and also Lelieveld and Crutzen, 1991; Jiang et al., 1997; Barth et al., 2003; Ervens et al., 2008, and Tilgner et al., 2010.
6. Section 6.1. More information about the case needs to be included. Specifically, the latitude, longitude, altitude, and size of the drops. Later in the discussion of the results, "sunset" is often used but what time is sunset?

7.  Section 6. I realize that the test case is an example. However, air parcels do not spend 12 hours in a cloud. Vertical motions maintain a cloud by moving air above its lifting condensation level. Thus, air parcels are constantly being transported into the cloud region (and out), and residence times are on the order of 10 minutes to 60 minutes. I suggest adding a statement commenting on this caveat. It is important because the results from this test case show that after a few hours aqueous chemistry controls the concentration of a dissolved trace gas, but in reality that air parcel is not in the cloud after a few hours.

8.  Page 14, line 1. The isoprene diurnal profile is not realistic. It is explained by the gas-phase chemistry. Another factor is that the isoprene emissions are constant with time, while in reality they vary diurnally. Please discuss this factor in the paper.

9.  Page 14, lines 9-12. I agree with the explanation of the $H_2O_2$ time evolution. In addition, $SO_2$ concentrations must be depleted in order for $H_2O_2$ concentrations to increase. Otherwise $H_2O_2$ would continue to be consumed by $SO_2$. Further, it is worth noting the time scales with respect to how long an air parcel actually spends inside a cloud.

10. Page 15, lines 16-19. How long does it take for aqueous-phase glyoxal concentrations to change from being controlled by mass transfer to being controlled by aqueous chemistry? Should we expect to see this in observations of clouds in the atmosphere where the air parcel residence time in cloud may be shorter?

11. Page 15, lines 22-23. The comment about acetic and formic acid concentrations is interesting, but I did not see these values plotted. Could they be included in the figure (or at least report the concentrations)?

12. Page 15-16. I did not completely understand the importance of dissolved organic carbon (DOC) and its role in the aqueous chemistry. Could more background information be provided?

13. The supplementary material shows the pH values for the simulation. The pH seems to be quite low (3-3.5). Could the authors explain why such a low pH occurs? Are the results similar to previous simulations of this case?

Technical Comments

1.  Page 1. line 20. → multiphase
2.  Page 1, line 34. "GROMHE" is not defined. Is it needed in the abstract?
3.  Page 1, line 38. → The photolysis rates in both phases ….
4.  Page 1, line 39. The word "evaluate" is not what is done in the paper.
5.  Page 2, line 33. Please quantify what is meant by "low-$NO_x$" and "dilute".

6. Section 2. Please cite the supplementary material listing the chemical mechanism and chemical species. This supplementary material is a good resource for researchers.
7. Page 3, Line 6. Please quantify "significantly soluble and highly reactive". For example, I would suggest saying that their Henry's Law coefficient is greater than a specific value.
8. Page 3, line 28. Please add information on what kind of "data are available". I assume that Henry's Law coefficients and reaction rates are meant, but this needs to be clear.
9. Page 3, line 31. Define "GROMHE".
10. Page 5, line 23. Could a couple of sentences be added to explain "global reaction rate constants"?
11. Page 6, line 22. → Such a large set of species
12. Page 6, line 23. It seems that this paragraph should be part of the previous paragraph.
13. Page 8, line 14. → Although these reaction rates
14. Page 9, line 19. Add that the process is explained below this paragraph.
15. Page 11, line 20. → The last reaction
16. Page 12, line 13. Lelieveld and Crutzen (1991) adopted an accommodation coefficient value of 0.05 for soluble gases in which the accommodation coefficient was not known. Perhaps this reference began this practice and should be cited.
17. Page 13, line 20. → simulation has been run for 31 days. What is the purpose of a 31-day gas chemistry spin up?
18. Page 14, line 2. A sentence should be added to say something about the isoprene products diurnal profiles.
19. Page 14, line 16. When is sunset?
20. Page 14, line 22. → is also responsible
21. Page 14, lines 23-27. I suggest discussing the "with DOC" results together with the last paragraph on the page. That is, organize the discussion of Figures 3 and 4 to present "without DOC" results first, and then discuss "with DOC" results presented in Figures 3 and 4.
22. Page 14, line 32. When is nighttime?
23. Page 15, line 7. → in terms of concentrations
24. Page 15, line 13. In SM6, it would be helpful to either put names with the chemical formulas, or organize the list by groups (alkanes, alkenes, etc.) as is done on the MCM web page.
25. Page 15, line 25. → acids as main contributors
26. Page 15, line 25. Since this topic sentence has been known for a while, it would be good to cite the appropriate reference, for example Chameides (1984) JGR.
27. Page 15, line 37. → 15 LT   ...   14 LT  (local time)

28. Page 16, line 9. → protocol provides
29. Page 16, line 12. → introducing, for example, the
30. Page 16, line 13. Remove "compounds"
31. Page 16, line 17, → to impact the O/C ratio
32. Page 16, lines 33-37 may be better placed at line 21
33. Table 4. I did not see where the (b) footnote is cited. What does the "constant" refer to?
34. Figures: Could there be tick marks on every axis so that it is easy to locate where the time is for each concentration time evolution?
35. Figure 5: The legend connected with the bottom 2 panels is too small to read. It may not be needed if it is the same as the legend in the top 2 panels.

---

## Author Comment (AC1) · 10 Jan 2017

We would like to thank you for the note. We will add a version number (1.0) to the manuscript title in the revised submission of the manuscript.

---

## Author Comment (AC2) · 10 Jan 2017

**Response to Anonymous Referee #1**

**We thank Anonymous Referee #1 for the interesting comments on our manuscript. All the individual comments are addressed below in bold.**

**Main comments**

*1) Motivation of the current study*

*The intention of developing a 'complete' and 'correct' aqueous phase mechanism is clearly an ambitious and laudable project. However, it is clear that due to the lack of data many assumptions have to be made that should be better justified here. While such a mechanism can be used in a box model to assess chemical interactions in the multiphase system, it is impossible to track hundreds of species in a larger scale model. Therefore, I am missing a clearer direction on how this goal can be reached. For example, previous studies have suggested several strategies to reduce chemical mechanisms (Ervens et al., 2003; Ervens et al., 2008; Deguillaume et al., 2009; Woo and McNeill, 2015) for use in larger models. One way of reducing models is identifying the most important reactions and species. Such a discussion is missing here. – Instead as further directions it is suggested to include more species (succinic acid, tartronic acid).*

**Given that an almost complete MCM mechanism (4642 species and 13,566 reactions) has already been used in the CMAQ regional model (Ying and Li, 2011; Li et al., 2015), it is not technically impossible to implement the CLEPS mechanism coupled to MCM in a 3D model. However, the objective of this work is not to propose a chemical scheme for 3D models. Its goal is to provide a set of systematic rules to build nearly explicit mechanisms. In this scope, including more species is a way to test these rules and work toward their completeness. The matter of reducing chemical mechanisms to use them in large scale models is another (very complex) task. It is however not, in our opinion, in the scope of this paper.**

**The following was added to the conclusion to outline a use of CLEPS for developing reduced schemes:**

**"Several strategies can be used to build reduced multiphase mechanism. Explicit schemes like CLEPS can serve as a reference for the development of reduced chemical schemes like those proposed in Woo and McNeill (2015) or Deguillaume et al. (2009). Very few experimental data are available to validate reduced chemical scheme before implementation in 3D models, especially concerning cloud chemistry. Field measurements (van Pinxteren et al., 2015) and chamber experimental data (Bregonzio-Rozier et al., 2016) are available, but they do not cover the whole range of real atmospheric conditions that a large scale model can encounter. Using explicit modelling tools to serve as reference for a large set of environmental and chemical scenarios is therefore a possible solution to validate reduced multiphase chemical schemes for various environmental conditions. This is of course consistent only once the explicit mechanisms have been validated against the existing experimental data."**

*2) Similarities to CAPRAM*

*a) In the abstract and also later, it is claimed that 'a new detailed aqueous phase mechanism . . . is proposed'. Given that all inorganic chemistry seems the same as other mechanisms (such as CAPRAM) and also all 'overall' rate constants are the same, I think this statement is highly exaggerated. I suggest marking in the reaction tables those processes and parameters that are new or different.*

**We don't think it is highly exaggerated to claim that the mechanism is new. There are 207 inorganic reactions and 642 organic reactions. The inorganic reactions represent 24% of the full scheme. As the literature data is scarce on the subject, CLEPS and CAPRAM obviously share a lot of experimental data concerning inorganic chemistry and $C_{1-2}$ chemistry.**

**In our opinion, a detailed comparison of this mechanism with the CAPRAM mechanism is not in the scope of this paper. This work has been submitted to GMDD to present the new CLEPS mechanism and the associated detailed protocol rules. We are confident that having several detailed mechanisms available to the scientific community, based on different assumptions, can only be beneficial to better understand the processes governing aqueous phase chemistry. The confrontation between CLEPS and CAPRAM will be the subject of further studies to evaluate for instance the sensitivity of modelled cloud systems to the assumption used to build multiphase mechanisms.**

**The following have been added in different places in the paper to underline differences with CAPRAM:**

[revised manuscript text omitted]

*b) It seems to me that several simplifications that have been tested in CAPRAM (Ervens et al., 2003) are repeated here – without the previous detailed sensitivity studies. Such simplifications include skipping the formation of peroxy radicals as an individual step (p. 9, l. 1-5), the self-recombination of peroxy radicals (p. 9, l. 10) and skipping of the tetroxide (p. 9-10). The previous work should be properly referenced here.*

**This has been addressed in the previous answer, in the added sentences to sect. 4.1 and 4.2.**

*c) How does the analysis of OH(aq) sinks and sources (p. 15, top) compare to previous studies such as CAPRAM (Herrmann et al., 2000; Ervens et al., 2003; Deguillaume et al., 2009; Tilgner et al., 2013)?*

In Tilgner et al. (2013), HO$^\bullet$ sources in cloud for a remote scenario similar to our simulation are dominated by direct uptake from the gas phase, by the Fenton type reactions and by the photolysis of $H_2O_2$ and Fe(III). Sinks are dominated by reactions with organic compounds, especially formate, hydrated formaldehyde, glycolaldehyde and methylglyoxal.

The following was added in the corresponding paragraph:

"However, in the first hours of the cloud event, mass transfer is the major source of HO$^\bullet$, like it was predicted in a previous modeling study on a shorter cloud event considering a remote chemical scenario (Tilgner et al., 2013). Fenton type reactions and photolysis reaction are also significant sources of HO$^\bullet$ in their simulation." and "Tilgner et al. (2013) also show that HO$^\bullet$ only aqueous sinks are reactions with organic matter, especially carbonyl compounds such as hydrated formaldehyde, glycolaldehyde and methylglyoxal."

*d) How does the modeled evolution of the O/C ratio compare to the trends as shown by (Schrödner et al., 2014)? Can the differences be ascribed to the more complex formulation of the branching ratios of the organic reactions?*

The O/C trends shown in Schrödner et al. (2014) have been calculated for organic aerosol (figure 10 in the article). They have shown higher values than those modelled in our work for the O/C ratio in the aqueous phase. For instance in their simulated clouds, modelled O/C ratios reach 1.7 after 48h in the rural cases, while in our simulations O/C ratios reach 1.1 values after 12h. This difference might be ascribed to higher reactive organic mass and lower HO$^\bullet$ concentrations considered in our model. For instance, our test case considering DOC reactivity increases the reactive organic mass from approx. 5 $\mu$g m$^{-3}$ to approx. 20 $\mu$g m$^{-3}$. This has the effect of reducing HO$^\bullet$ concentration and O/C ratios. In Schrödner et al. (2014) simulations, organic mass reaches 3.1 $\mu$g m$^{-3}$ in the lower cloud, which can be the reason why they model higher O/C ratios. This is highly hypothetical, as the modelling setup and studied scenarios in Schrödner et al. (2014) are strongly different from the model runs in this work. To our opinion, we don't believe that mentioning this comparison is pertinent in this paper.

*3) Basis of assumptions*

*a) Assumptions on equilibria (hydration, dissociation, gas/aqueous partitioning)*

*In Sections 3.1, 3.2 and 5.2, it is explained how unknown equilibrium constants are estimated. While it is clearly necessary to estimate such parameters due to the lack of their availability, it should be shown (i) the validity of these assumptions, (ii) a comparison of the estimated parameters to (the few) known parameters and (iii) which assumptions have been made in other aqueous phase mechanisms and whether these assumptions are less appropriate.*

Hydration equilibrium constants estimated with the GROMHE method have been evaluated in Raventos-Duran et al. (2010). Our modification concerning carboxylate ions is already discussed in Sect. 3.1 and in Fig. 1. We acknowledge that our estimates of acidity constants are a simple first approach. Further work should look into the validity of our assumptions on this subject, for example considering the work of Perrin et al. (1981). For all these assumptions, sensitivity studies should be carried out to check if our approximations need to be refined or not. In our opinion, this work is not in the scope of this paper.

**Please refer to our answer to comment 2a) for comparison to the assumptions made in the CAPRAM mechanism.**

*b) I understand that a completely consistent coupling of MCM and the aqueous phase mechanism is difficult. However, the fact that numerous species do not have a gas or aqueous phase equivalent, respectively (p. 11, Section 5.1) introduces an unnecessary inconsistency. In my opinion, it would be more consistent to introduce a solubility threshold (Henry's law constant) that determines where a species should reside.*

**The following has been added to sect. 5.1 to clarify our proposed solution for this inconsistency.**

**"For each species with no equivalent in the other phase, we create an artificial equivalent in the other phase for which no reactivity is described. The mass transfer parameters are estimated as described below (section 5.2) to accurately determine in which phase the species should reside."**

*c) How well do calculated Henry's law constants based on GROMHE SAR compare to measured values? Can you refer to some comparison in a previous study or show it here (possibly in supplemental information)?*

**The calculated Henry's law constants based on GROMHE SAR have been thoroughly evaluated in Raventos-Duran et al. (2010). The following sentences have been added in sect. 5.2 for more consistency:**

**"Comparing this SAR with other available methods (Meyland and Howard, 2000; Hilal et al., 2008) Raventos-Duran et al. (2010) have shown that GROMHE is the more reliable SAR in general, estimating Henry's law constants with a root mean square error of 0.38 log units (approx. a factor of two). It particularly shows better performances than the other tested methods for the more soluble species, *i.e.* highly oxygenated, multifunctional organic species."**

*4) Inconsistencies in the presented mechanism*

*a) Dissociation equilibria*

*- In Eq. 4, the resulting $K_A$ constant should be dimensionless. In the Equilibria table in the supplemental information, data are given in M (according to literature). The unit is missing in this table and therefore it is very confusing and leads to a bias of a factor 18 (molecular weight of water) in all dissociation constants. How have the numbers in the table been used in the model?*

*- Later in the manuscript (p. 12, l. 28), it is stated that the equilibria are split into forward and back reaction. What values are used? How were they derived?*

**The units have been clarified in the tables: $K_A$ values are in M and $K_h$ values are dimensionless. Eq. 4 and R.2 have been rewritten to match the data.**

**R.2 >CO(OH) ↔ >CO(O⁻) + H⁺**

**Eq 4. $K_A = \dfrac{[>CO(O^-)][H^+]}{[>CO(OH)]}$**

**In the model, equilibrium constants are used to derive back and forward reactions, based on the assumption that the equilibria are quasi-instantaneous. The forward rate constant $k_f$ [s⁻¹] is assumed to be very fast, with for instance:**

**$k_f = 10^6$ s⁻¹**

**To reach equilibrium, the backward rate constant must then be:**

$$k_b = k_f/K_A$$

**Because $k_f$ is considered to be fast, implementing equilibria in this way yields the same results as implementing a total species approach which assume that equilibria are instantaneous. In this work, we provide equilibrium constants, and the implementation details should vary from a boxmodel to another. Future work should be carried out to study the sensitivity of the system to the chosen $k_f$. Known acid dissociation rate constants are in the range of $10^6$-$10^8$ s$^{-1}$ (see for example the constant listed in CAPRAM 2.4; Ervens et al., 2003). For such high rate constants, the choice of $k_f$ shouldn't be sensitive. This is bigger issue for hydration which is known to be slower and pH dependent (see for instance the discussion in Doussin and Monod, 2013).**

*b) Table 2 I do not understand the third example in Table 2. The 'global rate constant' is given as $3.2e8$ $M^{-1}$ $s^{-1}$; however, all rate constants for the various branching pathways exceed this number and yield a value that is an order of magnitude greater. Is this just a typo or were these (and possibly other?) rate constants used like this used in the model?*

**In the given example, $k_1 = 8.1 \times 10^7$ M$^{-1}$ s$^{-1}$, $k_2 = 6.1 \times 10^7$ M$^{-1}$ s$^{-1}$, $k_3 = 8.5 \times 10^7$ M$^{-1}$ s$^{-1}$, $k_4 = 9.2 \times 10^7$ M$^{-1}$ s$^{-1}$ : $k_1 + k_2 + k_3 + k_4 = 3.2 \times 10^8$ M$^{-1}$ s$^{-1}$. Therefore, the $3.2 \times 10^8$ M$^{-1}$ s$^{-1}$ value is not a typo. However, there is a real typo in the table for the final estimated global reaction rate ($3.2 \times 10^9$ instead of $3.2 \times 10^8$) of the same species. We suppressed this mistake in the manuscript.**

*5) Discussion of results*

*a) Length of cloud In several previous model studies, it has been discussed in detail that cloud processing time is on a time scale on the order of a few minutes or at most an hour as it is restricted by the lifetime of a droplet (e.g., Feingold and Kreidenweis, 2002; Ervens et al., 2004; Tilgner et al., 2013). Therefore, the conclusions of the aqueous phase on the oxidation capacity of the atmosphere should be revised and related to more atmospherically relevant conditions.*

**We are aware that our simulation is performed using rough microphysical cloud properties (constant LWC and radius) over a long cloud period. Our simulation was performed without considering microphysical processes that are needed to conduct realistic cloud simulations. These processes are included in a module which is about to be completed and the results of more realistic cloud simulations should be submitted soon to GMDD. In this work we use a permanent cloud because our primary goal is to verify that our mechanisms have an effect on the air-water system. In this work, we are able to show that our mechanism effects are visible for long term simulations. Moreover, we have analysed the detailed budget (sinks and sources) of chemical species and we also clearly show that cloud chemical reactivity is highly variable over time and that chemical species concentrations can vary significantly over short time scale (few minutes).**

**This was added in Sect. 6.1 to clarify this point: "This is a permanent cloud simulation and no attempt is made to represent a specific documented cloudy situation. The objective is to test the multiphase mechanism over a long time scale to check that the mechanism is (i) working as intended and (ii) producing chemical effects in both phases. Testing the model over 12h is a first step to evaluate the impacts (or their absence) of detailed organic chemistry on multiphase cloud chemistry. Future studies will use variable environmental conditions that require the consideration of microphysical processes with our multiphase chemical module."**

**Following this general comment, the conclusion was modified with the following sentences: "These simulations were conducted for a long non-realistic permanent cloud. However, the mentioned results are atmospherically relevant, since the impact on O/C ratio and fragmentation can be observed in the first moments of the simulated cloud event."**

*b) OH profiles: Several previous studies have shown that the OH concentration in the gas phase is substantially reduced in the presence of clouds (Herrmann et al., 2000;Ervens et al., 2003; Tilgner et al., 2013). The current mechanism shows a doubling of OH(gas) during cloud. Admittedly, the initial conditions were slightly different than in the CAPRAM simulations. However, given the robustness of the previous results for various scenarios, the differences of the current study to previous ones should be discussed.*

**This is now added in the discussion of Fig. 3a&3b in Sect. 6.3:**

**"This trend in HO˙ mixing ratios contradicts previous modeling results (Herrmann et al., 2000; Barth et al., 2003; Ervens et al., 2003; Tilgner et al., 2013) which exhibit a decrease in HO˙ mixing ratios during cloud events. The chosen chemical scenario might be the reason for this difference. Even if the chemical scenario in our study is rather similar to the one in Ervens et al. (2003), we still differ in the amount of emitted organic compounds. In our test simulations, we mainly emit isoprene, with a small contribution of formaldehyde and acetaldehyde, whereas Ervens et al. (2003) emit a larger range of hydrocarbons of anthropogenic (alkanes, alkenes, aromatics) and biogenic origin (limonene, α-pinene). As far as we understand the CAPRAM model setup, these hydrocarbons are not dissolved and it should be noted that they are highly reactive with HO˙. This means that the large, and certainly major, sink of gaseous HO˙ caused by hydrocarbons reactivity is always present, even under cloud conditions. When the source of HO˙ radicals is reduced by the cloud event (*e.g.* due to HO$_2$ and NO separation), HO˙ radicals sinks are not significantly perturbed and HO˙ steady state mixing ratios decrease. Conversely, in our simulation the gaseous HO˙ sink is more significantly perturbed by the cloud event because most of the organic matter in our scenario is produced from isoprene oxidation and is readily soluble. In our case, it seems that the HO˙ gaseous source reduction is overcompensated by the reduction in HO˙ gaseous sinks. As a consequence, HO˙ steady state mixing ratios are higher during cloud events. This hypothesis especially highlights how the chosen chemical scenario and regime is important for simulation results and conclusions. Future work should therefore systematically explore cloud simulations under a large range of scenarios."**

*c) How do the differences between the three scenarios in Fig. 3b compare to previous model studies? Can possible differences be ascribed to the new organic pathways?*

**The production of acids and destruction of carbonyl species in the gas phase *via* aqueous phase reactivity is similar to what has been shown in Tilgner et al. (2010) on smaller timescales. Differences in isoprene can be linked back to the reasons invoked in the previous answer.**

*6) Previous literature on aqueous phase chemisry*

*- p. 2, l. 12: In previous literature (Herrmann et al., 2015, and references therein) it is discussed that the presence of an aqueous phase (cloud) leads to the separation of the soluble HO$_2$ and the rather insoluble NO in the gas phase. Since their reaction in suppressed, less OH is formed. Therefore, it is not radical chemistry in the aqueous phase but different pathway strengths in the gas phase that lead to differences in OH during cloud.*

**This is noted and corrected.**

*- p. 2, l. 3: The statement that photolysis rates are 'highly enhanced in clouds' is not always true. While in thin clouds, photolysis rates are enhanced, in dense cloud they might be reduced (cf e.g. Fig. 2 in Ervens, 2015).*

**In this particular sentence, we are referring to photolysis inside cloud droplets and not inside interstitial air. It seems to us that the general consensus is that, depending on the position of a droplet in the cloud, the photolysis rates will be enhanced or reduced. However, theoretical calculations from Ruggaber et al. (1997) have shown that on average, actinic fluxes are indeed enhanced in cloud droplets.**

*- p. 3, l. 5: In the cited paper (Ervens et al., 2015) it is discussed that oligomerization of MACR and MVK is rather unimportant in the atmosphere because of the low solubility of these precursors.*

**We indeed overlooked the discussion about salting-out effects. The following sentence has been added: "Ervens et al. (2015) however argued in a modeling study that MVK and MACR solubility could be decreased by salting out effects, reducing their contributions to aqueous reactivity and SOA formation."**

*- p. 6, l. 30: How well does the product distribution estimated here matches the detailed laboratory study (Perri et al., 2009)?*

**Perri et al. (2009) show that glyoxal, formic acid, glyoxylic acid, glycolic acid, oxalic acid, malonic acid and succinic acid could be formed from HO$^{\bullet}$ oxidation of glycolaldehyde. In our mechanism, HO$^{\bullet}$ oxidation of glycolaldehyde directly produces glyoxylic acid, glyoxal, formic acid and formaldehyde. Oxalic acid is not a first generation product of the oxidation of glycolaldehyde, but it can be formed indirectly from the oxidation of glyoxal and glyoxylic acid. Accretion reactions are not included in our protocol but could be responsible for the production of succinic and malonic acids. However, it should be noted that Perri et al. (2009) experiments were carried out at glycolaldehyde concentrations 1000 times higher than typical clouds levels, which could favour the formation of heavier weight species.**

*- p. 8, Section 3.6: Results of a recent key paper on aqueous photolysis should be discussed here (Epstein et al., 2013).*

**We added the following in Sect. 3.6:**

**"Epstein et al. (2013) have shown that aqueous photolysis quantum yields are highly dependent on the type of molecule. Using similarity criteria to estimate photolysis rates in the aqueous phase may be too error prone. Furthermore their estimates also show that photolysis would efficiently compete with HO$^{\bullet}$ oxidation for very few of photolabile species. If more data and reliable SAR become available on this subject, a mechanism generated using the present protocol would be the ideal tool to expand on Epstein et al. (2013) study."**

*- p. 8, l. 34-36: In the study by Ervens et al., 2015 it is stated "Unlike in laboratory experiments, atmospheric aqueous aerosol particles can be considered saturated with oxygen (~270 μM) due to their large surface–volume ratio. In all our model sensitivity studies with the multiphase model, the oxygen concentration reached saturation level after a few seconds." – which is not clearly reflected in the text in the current manuscript.*

In the same paragraph, the sentence "In our mechanism, which is characterised by dilute conditions, dissolved oxygen is assumed to be always available (see Ervens et al., 2015) and alkyl radicals are too dilute to react with themselves." has been replaced with:

"In their bulk aqueous phase modeling study, Ervens et al. (2015) have shown that the aqueous phase under laboratory experiment conditions is not saturated with oxygen, leading to possible oligomerization. Their sensitivity studies however show that oxygen reached saturation in few seconds for atmospheric deliquescent particles, likely because of a large surface to volume ratio. We follow the same hypothesis for cloud droplets."

*- p. 14, l. 37ff: A lower OH concentration will not only lead to lower formation rates of these acids but also to lower destruction rates (Ervens et al., 2014). Therefore the role of OH for the total organic acid levels depends on the ratio of k(formation) /k(destruction), which, in turn, might be a function of pH. This should be mentioned here.*

This point is really interesting. However, our results show that pH doesn't change with added DOC. Therefore, differences in acids concentrations cannot be ascribed to asymmetrical reactivity between acids and corresponding bases. The only other possibility that we can think of are "fixed" sinks and sources not dependent on OH concentrations, *i.e.* photolysis and mass transfer.

This is now mentioned in Sect. 6.3.

"Because sinks and sources of acids due to HO$^{\bullet}$ radicals should vary in equal proportions, the decrease in organic acids concentrations cannot be ascribed to reactivity with HO$^{\bullet}$ radicals. We therefore has to consider fixed sinks that do not depend on HO$^{\bullet}$ concentrations, *i.e.* photolysis and phase transfer. If we consider that acids reach pseudo steady state concentrations, we can assume that because photolysis and phase transfer are not modified by the additional DOC, some acids concentrations could decrease following their overall sources/sinks ratio.".

**Technical and minor comments**

*p. 5, l. 21: 'Rate constants'*

Done.

*p. 7, l. 24; p. 8, l. 14, and some other places in the manuscript: 'Reaction rates' are defined as the product of a rate constant and reactant concentration(s). It should be 'rate constant'.*

Done.

*p. 10, l. 28: Give a reference for this concentration.*

This concentration comes from our simulated peroxyl radicals concentrations. The sentence has been replaced by the following: "In our simulation (see Section 6) the high range of concentrations for peroxyl radicals is around $10^{-10}$ M for the peroxyl radical derived from glyoxal; tetroxide formation therefore occurs on a timescale of approximately 50 s."

*p. 15, l. 26: Which acids are included in 'total acids'? Do they include formic and acetic acid?*

'Total acids' include formic and acetic acids. It has been clarified in the text.

*Table 1: The references below the table should be added to the main reference list and not listed here.*

**Done.**

*Table 4: What parameter does the footnote (b) refer to?*

**This footnote comes from a previous version of the paper. It is not used anymore and has been removed.**

*Figure 2, caption: 'Relative' should be replace by 'related'*

**Done.**

*Fig. 3 and 4: Add in the caption that the red line is hidden by the blue one or even choose different line types (e.g., dotted vs solid)*

**Done.**

*Fig. 5: Clarify the caption: '10 most important species' in terms of what?*

**This has been clarified. These species are the most important in terms of concentrations in the aqueous phase.**

---

## Author Comment (AC3) · 10 Jan 2017

**Response to Anonymous Referee #2**

**We thank Anonymous Referee #2 for the interesting comments on our manuscript. All the individual comments are addressed below.**

*This paper introduces a new protocol for developing a detailed aqueous chemistry mechanism. It emphasizes oxidation by OH, NO₃, and other oxidants in low NOₓ and dilute conditions, and does not include accretion processes. The detailed aqueous chemistry mechanism, coupled with the Master Chemical Mechanism (MCM v.3.3.1), is applied to an ideal cloud situation to examine the behavior of the chemistry.*

*This is an exciting, new advancement for our understanding of aqueous chemistry, especially for the oxidation of organic compounds dissolved in cloud water. The paper is fairly well written, although several spots need to be clarified. The paper can be improved by providing more information on the test case, giving more discussion on how the mechanism and results compare to previous studies (with and without detailed aqueous chemistry), and cleaning up the presentation of the paper. My suggestions of needed clarifications are given below.*

**Specific Comments**

*1. Page 6, line 25 and Table 2. It is not completely clear to me what criteria are used to proceed with the reduction scheme. For example, why is a branch with a 19% contribution to the reaction removed? Is it simply because the other 3 pathways represent >75% of the global reaction rate? It seems that pathways that comprise 10% or less of the reaction rate can be ignored without much impact on the overall reaction scheme. What kind of impact occurs when the pathway contribution is larger, such as the 19% for the third example in Table 2?*

**The criteria used to reduce the size of the chemical mechanism have been chosen to consider the main chemical pathways and to avoid the consideration of negligible products. The example chosen to illustrate the reduction strategy in the paper is interesting because for 2-hydroxy, 3-oxobutanoate, the four available pathways are almost equiprobable. However, this is a really rare case. For most of the reactions described in the mechanism, we only neglect 10 to 15% of the total species reactivity (see footnotes in the mechanism tables - supplementary material). We believe that the impact of this reduction strategy can only be properly assessed with automatically generated chemical mechanisms (Aumont et al., 2005; Mouchel-Vallon et al., 2013).**

*2. Section 5. How does the CLEPS mechanism compare to previous aqueous chemistry studies (e.g. CAPRAM)?*

**This part of the paper deals with the coupling of CLEPS with a gas phase explicit mechanism. We refer the reviewer to our answer to Reviewer 1's comment 2a, specifically our amendments to part 5.2 discussing the difference in constructing the mass transfer coupling between our protocol and CAPRAM.**

*3. Page 13, line 10. While Rosenbrock solvers have become commonplace in chemistry transport models, other solver techniques also adequately solve the gas-aqueous chemistry mechanism (e.g., Ervens et al., 2003, JGR; Barth et al., 2003, JGR; McNeill et al., 2012, ES&T).*

**We agree with the reviewer's comment. We modified the sentence as follows: "Differential equations are solved with a Rosenbrock solver which has been shown to be a reliable numerical method for stiff ODE systems involved in modelling multiphase chemistry".**

*4. Page 13, line 12. Please restate the objective of section 6. The introduction says that "the box model is tested for an ideal cloud situation", which implies some kind of evaluation. However, there is no evaluation of the model results provided and there is only one instance of a comparison with previous modeling work (page 14, line 30).*

*As a follow up to the previous comment, please include more discussion in Section 6 of how the model results in this paper compare to previous ones, e.g. McNeill et al., 2012; Tilgner et al., 2013; Herrmann et al., 2005 – which can hopefully be directly compared due to their similarity of the test case conditions, and also Lelieveld and Crutzen, 1991; Jiang et al., 1997; Barth et al., 2003; Ervens et al., 2008, and Tilgner et al., 2010.*

**We understand the reviewer's concern about comparisons between CLEPS and previous mechanisms. Differences in chosen physical and chemical scenarios make this exercise particularly perilous. On this subject, see for instance our answer to Reviewer 1 comment 5b. We believe that specific inter-comparison studies should be run to properly evaluate and compare existing mechanisms. This will be done in the future. However, this is not the scope of this particular paper. This paper aims at describing and presenting a detailed systematic protocol for aqueous phase oxidation. Hence, the mechanism might be viewed as the "main result" of this work. Nevertheless, qualitative comparisons with simulations from existing mechanism are addressed to some extent in our answers to Reviewer 1 comments 2b, 2c and in our new additions to Sect. 6.3., where some of our model outputs are compared to previous modelling studies using CAPRAM mechanism.**

*6. Section 6.1. More information about the case needs to be included. Specifically, the latitude, longitude, altitude, and size of the drops. Later in the discussion of the results, "sunset" is often used but what time is sunset?*

**This is noted and has been indicated in Sect. 6.1:**
**"The simulation is located at the sea level and the coordinates used to calculate actinic fluxes are 45.77°N 2.96°E."**
**Sunset time (6:45pm) is now included in Sect. 6.1.**

*7. Section 6. I realize that the test case is an example. However, air parcels do not spend 12 hours in a cloud. Vertical motions maintain a cloud by moving air above its lifting condensation level. Thus, air parcels are constantly being transported into the cloud region (and out), and residence times are on the order of 10 minutes to 60 minutes. I suggest adding a statement commenting on this caveat. It is important because the results from this test case show that after a few hours aqueous chemistry controls the concentration of a dissolved trace gas, but in reality that air parcel is not in the cloud after a few hours.*

**Reviewer 1 raised the same issue. We hope to have addressed it in our answer to Reviewer 1 comment 5a) and in the related modifications to the conclusion and sect. 6.1.**

*8. Page 14, line 1. The isoprene diurnal profile is not realistic. It is explained by the gas-phase chemistry. Another factor is that the isoprene emissions are constant with time, while in reality they vary diurnally. Please discuss this factor in the paper.*

**This problem has been addressed in Sect. 6.2: "The resulting isoprene diurnal profile is not realistic, as in the atmosphere the isoprene diurnal profile is constrained by the diurnal variation of both its emissions and level of oxidants."**

*9. Page 14, lines 9-12. I agree with the explanation of the $H_2O_2$ time evolution. In addition, $SO_2$ concentrations must be depleted in order for $H_2O_2$ concentrations to increase. Otherwise $H_2O_2$ would continue to be consumed by $SO_2$. Further, it is worth noting the time scales with respect to how long an air parcel actually spends inside a cloud.*

**In the simulations, sulphur is indeed quickly depleted. The end of the first paragraph of Sect. 6.3 has been modified as follows: "$H_2O_2$ is a soluble species highly reactive with $SO_2$, which explains the initial dip in its mixing ratio. After $SO_2$ is entirely depleted (not shown), the aqueous production of $H_2O_2$ is responsible for its subsequent higher gaseous levels."**

*10. Page 15, lines 16-19. How long does it take for aqueous-phase glyoxal concentrations to change from being controlled by mass transfer to being controlled by aqueous chemistry? Should we expect to see this in observations of clouds in the atmosphere where the air parcel residence time in cloud may be shorter?*

Our detailed analysis of production/destruction fluxes of aqueous glyoxal (not shown) shows that dissolution of gaseous glyoxal dominates aqueous glyoxal production at the start of the cloud event. After approx. 10 minutes, aqueous phase chemistry becomes the main source of glyoxal. It is difficult to conclude if this could be observed under natural conditions. It depends on droplet lifetime and air parcel residence time. Sensitivity studies are needed to further address this issue.

*11. Page 15, lines 22-23. The comment about acetic and formic acid concentrations is interesting, but I did not see these values plotted. Could they be included in the figure (or at least report the concentrations)?*
**We add in the text the reference to Deguillaume et al. (2014) that reports the mean concentrations of formic acid and acetic acid in cloud aqueous phase at the puy de Dôme station for the last ten years. In their classification for non-polluted conditions, the concentration of formic acid is on average equal to 6.3 µM (range : 0.8-29 µM) in comparison to the 4 µM of formic acid simulated by our model. The concentration of acetic acid is on average equal to 4.9 µM (range : 0-23 µM) in comparison to the 1.5 µM of acetic acid simulated by our model.**

*12. Page 15-16. I did not completely understand the importance of dissolved organic carbon (DOC) and its role in the aqueous chemistry. Could more background information be provided?*
**This importance of DOC as an $HO^\bullet$ scavenger has been for instance discussed by Arakaki et al. (2013). It arises from the observation that cloud sampling field studies are able to identify only a few percent of the total Dissolved Organic Content (Deguillaume et al., 2014; Herckes et al., 2013). Most aqueous phase mechanisms, including CLEPS, only describe the reactivity of this known minor fraction. What is called DOC is the major constituent of dissolved organic matter, for which very little is known. Using experimental data, especially total $HO^\bullet$ reactivity measurement, Arakaki et al. (2013) were able to propose an average DOC + $HO^\bullet$ rate constant.**
**We are able to reproduce a similar behaviour in our model. Because the reactivity of most species dissolved from MCM is not described in CLEPS, a large amount of the modeled dissolved organic content is not reacting toward $HO^\bullet$ in the aqueous phase. The "with DOC" version of the simulations addresses this by assuming that the unreactive content of the aqueous phase reacts with $HO^\bullet$ with an average DOC + $HO^\bullet$ rate constant suggested by Arakaki et al. (2013).**

*13. The supplementary material shows the pH values for the simulation. The pH seems to be quite low (3-3.5). Could the authors explain why such a low pH occurs? Are the results similar to previous simulations of this case?*
**The simulated pH value is rather low but in the order of magnitude than was observed in other cloud chemistry model for remote conditions. For instance, Herrmann et al. (2005) and Deguillaume et al. (2004) for permanent cloud simulations showed values between 3 and 4. We are aware that this is not explained in the paper. This pH results from the balance between the production of strong inorganic acids (mainly sulfuric acid in our simulation) and the production of weak organic acids. Low pH values can therefore arise from (i) a lack of weak acids compared to other modelling results or (ii) an overestimation of acidity constants for organic acids.**

**Technical Comments**

*1. Page 1. line 20. → multiphase*
**Done.**
*2. Page 1, line 34. "GROMHE" is not defined. Is it needed in the abstract?*
**It is now defined in the text.**

*3. Page 1, line 38. → The photolysis rates in both phases ....*
**Done.**

*4. Page 1, line 39. The word "evaluate" is not what is done in the paper.*
**We replace "evaluate" by "test".**

*5. Page 2, line 33. Please quantify what is meant by "low-NOx" and "dilute".*
**This has been clarified for low-NO$_x$ conditions. The "dilute" part is discussed in details in the revised version of Sect. 4.1.**

*6. Section 2. Please cite the supplementary material listing the chemical mechanism and chemical species. This supplementary material is a good resource for researchers.*
**Done.**

*7. Page 3, Line 6. Please quantify "significantly soluble and highly reactive". For example, I would suggest saying that their Henry's Law coefficient is greater than a specific value.*
**This formulation is indeed vague and not always true. For instance, MVK and MACR have low solubility (H$_{MVK}$ ≈ 1e$^{-2}$ M atm$^{-1}$ and H$_{MACR}$ ≈ 1e$^{-1}$ M atm$^{-1}$). They are however highly reactive in the aqueous phase due to their double bonds and abundant enough in the gas phase to constitute a significant source of oxygenated compounds in the aqueous phase, *via* dissolution and reactivity. This sentence has been modified as follows: "For instance methylglyoxal (MGLY), glyoxal (GLY), acrolein (ACR), methacrolein (MACR) and methylvinylketone (MVK) are significantly soluble (Henry's law constant > 10$^3$ M atm$^{-1}$) and/or highly reactive in the aqueous phase (Ervens and Volkamer, 2010; Lim et al., 2010, 2013; Liu et al., 2009, 2012)."**

*8. Page 3, line 28. Please add information on what kind of "data are available". I assume that Henry's Law coefficients and reaction rates are meant, but this needs to be clear.*
**Done.**

*9. Page 3, line 31. Define "GROMHE".*
**Done.**

*10. Page 5, line 23. Could a couple of sentences be added to explain "global reaction rate constants"?*
**"global" reaction rate constants is used in opposition to partial reaction rate constants (*i.e.* site specific). This is a confusing formulation. The sentence has been replaced with: "When rate constants of organic compounds reactions with HO$^•$ are available (see the review from Herrmann et al., 2010), they are used in the mechanism."**

*11. Page 6, line 22. → Such a large set of species*
**Done.**

*12. Page 6, line 23. It seems that this paragraph should be part of the previous paragraph.*
**Done.**

*13. Page 8, line 14. → Although these reaction rates*
**Done.**

*14. Page 9, line 19. Add that the process is explained below this paragraph.*
**Done.**

*15. Page 11, line 20. → The last reaction*
**Done.**

*16. Page 12, line 13. Lelieveld and Crutzen (1991) adopted an accommodation coefficient value of 0.05 for soluble gases in which the accommodation coefficient was not known. Perhaps this reference began this practice and should be cited.*
**We added the reference from Lelieveld and Crutzen (1991).**

*17. Page 13, line 20. → simulation has been run for 31 days. What is the purpose of a 31-day gas chemistry spin up?*
**The purpose of a gas chemistry spin up is to obtain a diurnal steady state representative of the chose scenario. The duration of this spin up only has to be long enough to reach the pseudo steady**

state. It has been arbitrarily set to 31 days but it could have been shorter, as the steady state is reached after approx. 20 days.

*18. Page 14, line 2. A sentence should be added to say something about the isoprene products diurnal profiles.*
**The following has been added at the end of Sect 6.2: "The first oxidation products from isoprene (MACR, MVK) follow the same time profile as isoprene. The mixing ratios of other oxidation products vary also temporally depending on their production/destruction rates. For example, MGLY, GLY and glycolaldehyde mixing ratios decrease initially due to their oxidation by HO· and then increase strongly due to their production by the oxidation of isoprene."**

*19. Page 14, line 16. When is sunset?*
**In our simulation, the sunset corresponds to 6:45 PM. It has been added in the manuscript.**

*20. Page 14, line 22. → is also responsible*
**Done.**

*21. Page 14, lines 23-27. I suggest discussing the "with DOC" results together with the last paragraph on the page. That is, organize the discussion of Figures 3 and 4 to present "without DOC" results first, and then discuss "with DOC" results presented in Figures 3 and 4.*
**We respectfully disagree with this suggestion. The behaviour described for the "without DOC" simulation can often be highlighted with comparison to the "with DOC" simulation. In our opinion, it makes more sense to discuss these two simulations together, rather than separately.**

*22. Page 14, line 32. When is nighttime?*
**Nighttime corresponds to the time period from 6:45 PM et 12:00 PM. This has been added in the text.**

*23. Page 15, line 7. → in terms of concentrations*
**Done.**

*24. Page 15, line 13. In SM6, it would be helpful to either put names with the chemical formulas, or organize the list by groups (alkanes, alkenes, etc.) as is done on the MCM web page.*
**This is a difficult request to grant! We thought that the chemical formulas might be of some use to the potential reader. The list was automatically generated from our computer code. The order is arbitrary. Putting names and organising the list might not be worth the time needed to do it. We'll find a compromise for this problem for future versions of this mechanism.**

*25. Page 15, line 25. → acids as main contributors*
**Done.**

*26. Page 15, line 25. Since this topic sentence has been known for a while, it would be good to cite the appropriate reference, for example Chameides (1984) JGR.*
**Done.**

*27. Page 15, line 37. → 15 LT … 14 LT (local time)*
**Done.**

*28. Page 16, line 9. → protocol provides*
**Done.**

*29. Page 16, line 12. → introducing, for example, the*
**Done.**

*30. Page 16, line 13. Remove "compounds"*

**Done.**

*31. Page 16, line 17, → to impact the O/C ratio*
**Done.**

*32. Page 16, lines 33-37 may be better placed at line 21*
**We completed this paragraph following reviewer 1 comments. We prefer not to move this paragraph.**

*33. Table 4. I did not see where the (b) footnote is cited. What does the "constant" refer to?*
**This footnote comes from a previous version of the paper. It is not used anymore and has been removed. We apologize for this.**

*34. Figures: Could there be tick marks on every axis so that it is easy to locate where the time is for each concentration time evolution?*
**Figures have been made more readable.**

*35. Figure 5: The legend connected with the bottom 2 panels is too small to read. It may not be needed if it is the same as the legend in the top 2 panels.*
**Fixed.**

**Cited References**

Arakaki, T., Anastasio, C., Kuroki, Y., Nakajima, H., Okada, K., Kotani, Y., Handa, D., Azechi, S., Kimura, T., Tsuhako, A. and Miyagi, Y.: A general scavenging rate constant for reaction of hydroxyl radical with organic carbon in atmospheric waters., Environ. Sci. Technol., 47(15), 8196–203, doi:10.1021/es401927b, 2013.

Aumont, B., Szopa, S. and Madronich, S.: Modelling the evolution of organic carbon during its gas-phase tropospheric oxidation: development of an explicit model based on a self generating approach, Atmos. Chem. Phys., 5, 2497–2517, 2005.

Deguillaume, L., Leriche, M., Monod, A., Chaumerliac N.: The role of transition metal ions on HO$_x$ radicals in clouds: a numerical evaluation of its impact on multiphase chemistry, Atmos. Chem. Phys.,4, 95-110, 2004.Deguillaume, L., Charbouillot, T., Joly, M., Vaïtilingom, M., Parazols, M., Marinoni, A., Amato, P., Delort, A.-M., Vinatier, V., Flossmann, A., Chaumerliac, N., Pichon, J. M., Houdier, S., Laj, P., Sellegri, K., Colomb, A., Brigante, M. and Mailhot, G.: Classification of clouds sampled at the puy de Dôme (France) based on 10 yr of monitoring of their physicochemical properties, Atmos. Chem. Phys., 14(3), 1485–1506, doi:10.5194/acp-14-1485-2014, 2014.

Ervens, B. and Volkamer, R.: Glyoxal processing by aerosol multiphase chemistry: towards a kinetic modeling framework of secondary organic aerosol formation in aqueous particles, Atmos. Chem. Phys., 10(17), 8219–8244, 2010.

Herrmann, H., Tilgner, A., Barzaghi, P., Majdik, Z., Gligorovski, S., Poulain, L. and Monod, A.: Towards a more detailed description of tropospheric aqueous phase organic chemistry: CAPRAM 3.0, Atmos. Environ., 39(23–24), 4351–4363, doi:10.1016/j.atmosenv.2005.02.016, 2005.

Herrmann, H., Hoffmann, D., Schaefer, T., Bräuer, P. and Tilgner, A.: Tropospheric aqueous-phase free-radical chemistry: radical sources, spectra, reaction kinetics and prediction tools., Chemphyschem, 11(18), 3796–822, doi:10.1002/cphc.201000533, 2010.

Herckes, P., Valsaraj, K. T. and Collett, J. L.: A review of observations of organic matter in fogs and clouds: Origin, processing and fate, Atmos. Res., 132–133, 434–449, doi:10.1016/j.atmosres.2013.06.005, 2013.

Lelieveld, J. and Crutzen, P. J.: The role of coulds in tropospheric photochemistry, J. Atmos. Chem., 12(3), 229–267, 1991.Lim, Y. B., Tan, Y., Perri, M. J., Seitzinger, S. P. and Turpin, B. J.: Aqueous chemistry and its role in secondary organic aerosol (SOA) formation, Atmos. Chem. Phys., 10(21), 10521–10539, 2010.

Lim, Y. B., Tan, Y. and Turpin, B. J.: Chemical insights, explicit chemistry, and yields of secondary organic aerosol from OH radical oxidation of methylglyoxal and glyoxal in the aqueous phase, Atmos. Chem. Phys., 13(17), 8651–8667, doi:10.5194/acp-13-8651-2013, 2013.

Liu, Y., El Haddad, I., Scarfogliero, M., Nieto-Gligorovski, L., Temime-Roussel, B., Quivet, E., Marchand, N., Picquet-Varrault, B. and Monod, A.: In-cloud processes of methacrolein under simulated conditions - Part 1: Aqueous phase photooxidation, Atmos. Chem. Phys., 9(14), 5093–5105, 2009.

Liu, Y., Siekmann, F., Renard, P., El Zein, A., Salque, G., El Haddad, I., Temime-Roussel, B., Voisin, D., Thissen, R. and Monod, A.: Oligomer and SOA formation through aqueous phase photooxidation of methacrolein and methyl vinyl ketone, Atmos. Environ., 49, 123–129, doi:10.1016/j.atmosenv.2011.12.012, 2012.

Mouchel-Vallon, C., Bräuer, P., Camredon, M., Valorso, R., Madronich, S., Herrmann, H. and Aumont, B.: Explicit modeling of volatile organic compounds partitioning in the atmospheric aqueous phase, Atmos. Chem. Phys., 13(2), 1023–1037, doi:10.5194/acp-13-1023-2013, 2013.

---

## Author Response (AR2)

Dr Laurent Deguillaume

Dr. Camille Mouchel-Vallon

February 24, 2017

To whom it may concern,

Geoscientific Model Development

Copernicus Publications

Dear Sir, Madam,

Please find enclosed our corrected article submitted to Geoscientific Model Development (GMD). The article is entitled: *CLEPS: A New Protocol for Cloud Aqueous Phase Oxidation of VOC Mechanisms*, authored by C. Mouchel-Vallon, L. Deguillaume, A. Monod, H. Perroux, C. Rose, G. Ghigo, Y. Long, M. Leriche, B. Aumont, L. Patryl, P. Armand and N. Chaumerliac.

This article describes in detail a protocol defining rules to write explicit organic oxidation chemical mechanisms for cloud chemistry. It is based on an extensive literature review and relies on the latest developments in empirical structure-activity relationships. Based on this protocol a new cloud chemistry model is proposed with an explicit aqueous chemical mechanism (CLEPS) coupled to the Leeds Master Chemical Mechanism.

Following the second report of the reviewer, we proceed with the specific corrections and we provide detailed replies point by point to the comments. The points raised by the reviewer have significantly improve the paper. A new section has been completely rewritten to highlight the specificities of the CLEPS mechanism which can sometimes differ from CAPRAM (Ervens et al., 2003; Hermann et al., 2005) and to answer some of the reviewer comments dedicated to this particular comparison. Similarities and discrepancies between the two schemes are discussed and examples are provided for justifying choices in CLEPS when it is possible. Also, this new section allows avoiding repetitions and globally reduces the length of the paper as required by the reviewer. All the changes are indicated in a marked-up manuscript version for more clarity. We hope that the new revised version of the manuscript will be accepted for publication in GMD.

Sincerely,

Laurent Deguillaume

[Figure]

**Response to the reviewer:**

The manuscript has improved and the authors did a better job in explaining similarities to previous work. However, I still have several comments before the manuscript may be accepted for publication. The line numbers in my comments below refer to the marked-up manuscript that was attached to the response to the reviewers.

**First, we would like to thank the reviewer for his/her valuable comments. We are aware that some parts of the paper still needed more clarity. To improve this, a new section dedicated to the comparison of our mechanism CLEPS with CAPRAM has been added (section N° 5) in the manuscript. We also justify our chemical assumptions to better represent the oxidation processes of organic compounds in the aqueous phase.**

1) The abstract is rather lengthy, in particular in comparison to the introduction. I suggest shortening the abstract.
**Yes, we agree with this comment. The abstract has been shortened.**

2) In general, the paper is very long and tedious to read. I suggest trying to shorten it where possible. I point out some repetitive paragraphs below.
**We can understand the reviewer's feeling. We believe that the revisions discussed here have shortened it and that the addition of a new section helps removing repetitive paragraphs.**

3) While discussing the commonalities of CAPRAM and CLEPS, the authors make it sound as if there is only one set of rate constants for inorganic and many organic reactions. A look at the NIST data base 'Solution Kinetics' shows that for many reactions there are multiple values with sometimes great discrepancies. CAPRAM 2.4 was built and revised and includes recommended values. This should be noted here.
**CLEPS is also built upon a set of recommended data from the older versions of the mechanism (M2C2, see Leriche et al., 2000, 2003, 2007 ; Deguillaume et al., 2004; Long et al., 2013). It started on the basis of the Jacob (1986) mechanism and was progressively extended and updated to increase details about inorganic chemistry, to add C1, C2, TMI chemistry as well as oxalic-iron complexes formation and photo-oxidation. In the new Section 5, the following was added to clarify this point: "These two mechanism are built upon their own set of recommended data (*e,g.*, Ervens et al., 2004 for CAPRAM; Leriche et al., 2000, 2003, 2007; Deguillaume et al., 2004 for CLEPS)."**

4) In some parts, it still seems to me that important previous work was not considered. In a detailed analysis, it has been shown that the role of radicals other than OH for oxidation of organics in the aqueous phase is minor [Ervens et al., 2003a]. This should be at least mentioned and the addition of these radicals to the current mechanism justified. Has an analysis be done that shows that any reaction rates with other radicals that exceed those of OH?
**As mentioned in Section 3.1.4, $NO_3\cdot$ is the main night-time oxidant in the gas phase. In the low-$NO_x$ conditions used in our test-case simulation, its role is minor in both phases. However, Herrmann et al. (2010) mention $NO_3\cdot$ among the radicals that need to be considered for organic aqueous phase chemistry.**

**For instance, future studies may identify conditions for which NO₃˙ chemistry is important (*e.g.*, night time oxidation).**

**This question is already discussed in Section 3.1.4. We added a reference (Ervens et al., 2003a).**

5   5) Doussin and Monod were by far not the first ones who published SARs for OH reactions with organics. Many of the data in their SAR were taken from the data set that was used earlier for similar SARs [Ervens et al., 2003b; Herrmann, 2003].

**As mentioned in Section 3.1.3.3, Monod et al. (2005), Monod and Doussin (2008) and Doussin and Monod (2013) and simultaneously Minakata et al. (2009) are the first to publish SARs that allow estimating partial**
10  **H-abstraction rate constants. Moreover estimating branching ratios by means of these SARs is also possible, contrarily to the methods exposed in Ervens et al. (2003b) and Herrmann (2003).**

6) There are several SARs available for NO₃ and SO₄⁻ reactions. They should be referred to and discussed in 3.4.2, *etc.* [Herrmann, 2003]. These SARs should be used instead of omitting reactions for SO₄⁻ and NO₃ since
15  this leads to a bias in the treatment of organic sinks.

**The NO₃˙ and SO₄˙⁻ rate constants estimation methods presented in Herrmann (2003) or Hoffmann et al. (2009) rely on using measured Bond Dissociation Energies (BDEs), or on estimating them when they are not available. Since we estimate NO₃˙ reaction rates for all stable organic species in the CLEPS mechanism, the bias discussed by the reviewer can only be shown for SO₄˙⁻ reactions.**
20  **BDEs are most of the time not available for aqueous solutions. Therefore gas phase values are used, and Bensons incremental method for the gas phase is applied when needed (Ervens et al., 2003b ; Hoffmann et al., 2009). These correlations have not been used in the CLEPS mechanism because i) BDEs estimations are often accompanied by high uncertainties, and ii) the assumption behind the use of gas phase data to treat aqueous phase data is that the solvent effect on both reactants and products is energetically identical**
25  **for all reactions proceeding by H abstraction, which may be somewhat questionable in the case of acid (such as HNO₃ and HSO₄⁻) formation in water.**

**In the new section 5, we also give an example to demonstrate that CLEPS is more likely to take into account the variety of possible oxidation products that are measured in the lab: "For instance, in Table 2 the hydrated glycolaldehyde final reactivity in CLEPS is equally distributed between three HO˙ attack**
30  **sites and yields 33% glyoxylic acid, 28% glyoxal, 39% formic acid and formaldehyde. This result can be compared with the mechanism in CAPRAM 2.4 (Ervens et al., 2004) which leads to 100% glyoxylic acid since it only considers the aldehydic hydrogen abstraction."**

7) Now the authors point out at several places that the cloud period of 12 hours is not realistic. Based on these
35  statements, the 'test' of the model (p. 1, l. 40) should be better justified. Why not using shorter cloud periods of one hour or less as it has been done in previous model studies? Such tests do not require any sophisticated microphysics since it can be handled just like the performed model run by switching on the cloud for a realistic time after some initialization period. Given that the cloud is unrealistic, the comparisons to ambient measurements (p. 17, l. 34) does not seem very meaningful.

**This paper aimed at looking at aqueous phase chemistry and for this we used an idealized cloud scenario that have been used in many others studies (Jacob et al., 1986; Hermann et al., 2000; 2005,…). The coupling of the CLEPS model with a full microphysical model is the next step and at first will complexify the evaluation of the aqueous phase reactivity.**

5 **The new version of the paper now mentions the first hour of the cloud period, since this is indeed the period of time that can be meaningfully compared to ambient measurements: "In the first hours of the simulation, acetic and formic acids simulated concentrations are in the range of *in situ* measurements (Deguillaume et al., 2014)."**

*Herrmann, H., Ervens, B., Jacobi, H. W., Wolke, R., Nowacki, P., and Zellner, R.: CAPRAM2.3: A chemical*
10 *aqueous phase radical mechanism for tropospheric chemistry, J. Atmos. Chem., 36, 231-284, 10.1023/a:1006318622743, 2000.*

*Herrmann, H., Tilgner, A., Barzaghi, P., Majdik, Z., Gligorovski, S., Poulain, L. and Monod, A.: Towards a more detailed description of tropospheric aqueous phase organic chemistry: CAPRAM 3.0, Atmos. Environ., 39(23-24), 4351–4363, doi:10.1016/j.atmosenv.2005.02.016, 2005.*

15 *Jacob, D. J.: Chemistry of OH in remote clouds and its role in the production of formic acid and peroxymonosulfate, J. Geophys. Res., 91(D9), 9807–9826, 1986.*

8) Some more details on the O/C calculation (end of Section 6.3) should be given. The end result largely depends on the initial O/C ratio. It has been shown previously that the O/C ratio of background aerosol can be enhanced
20 due to cloud processing. This is not a new finding. Given the unrealistic cloud here, the resulting value is less important than the trend which has been shown by [Schrödner et al., 2014]. I do not agree with the authors that this reference is not relevant in this context. Also the statement in the conclusion about O/C ratio (p. 19, l. 8) should be discussed in the context of previous studies that have shown the same trends [Daumit et al., 2013; Ervens et al., 2014].

25 **The following sentence was added in Section 6.3 to clarify how O/C ratios are calculated and mention the modeled O/C ratios in Schrödner et al. (2014): "O/C is the ratio between the number of organic oxygen atoms and the organic carbon atoms in gas and cloud phases. The O/C ratios and $n_C$ are a measure of the extent to which long-chain organic species are oxidized and are therefore indicators of their functionalization and/or fragmentation. One hour after the start of aqueous phase chemistry, O/C in the**
30 **aqueous phase has remained around 1.0 and $n_C$ has decreased to 2.8 after a sharp initial increase to 2.9, thus showing that fragmentation is a major process. This result is in good agreement with other aqueous phase studies (Bregonzio-Rozier et al., 2016; Epstein and Nizkorodov, 2012; Epstein et al., 2013) and other models (Schrödner et al., 2014), but are in disagreement with field studies, probably due to a lack of descriptions of high molecular weight substances, and of their reactivity, as well as oligomerization**
35 **processes. The higher O/C ratios obtained by Schrödner et al. (2014) after cloud event (1.8 for their rural case) can be due to important oxalic acid concentrations dissolved into the aqueous phase in their model, when the cloud is being formed."**

**To our opinion, it is difficult to go into further comparisons with measured O/C ratios since most of the time the ratio relates to organic aerosol particles as in the study of Daumit et al. (2013) whereas in our**

**study the O/C in gas phase and in the cloud phase are discussed. In our study, the interest only lies in the fact that aqueous phase reactivity causes an increase in the O/C ratio in the gas phase especially when HO· is efficient (without DOC).**

**5    Minor comments**
* * *
- p. 2, l. 14: 'phase partitioning' should be replaced by 'phase separation'

**Yes, we replaced "phase partitioning" by "phase separation".**

- p. 2, l. 28: It is not true that CAPRAM v.3 is the most cited aqueous phase mechanism. The earlier version 2.3 has about twice as many citations (Web of Science) [Herrmann et al., 2000].

**Yes, we agree with the reviewer. CAPRAM 2.3 is the most cited version of CAPRAM. To our opinion, CAPRAM 2.3 is more related to inorganic chemistry and describes the reactivity of organic compound up to 2 carbon atoms. CAPRAM 3.0 (Herrmann et al., 2005) incorporates the former version CAPRAM 2.4 (Ervens et al., 2003) and a new extended reaction mechanism for atmospherically important organic compounds containing more than two and up to six carbon atoms. This mechanism is therefore more complete and is comparable to what is described in CLEPS. We rewrote the text to avoid any confusion.**

- p. 35: Define 'dilute conditions'.

**"Dilute conditions" is used to describe cloud droplets conditions, in opposition to deliquescent particles. This has been replaced by "cloud droplets".**

- p. 3, l. 17: MVK and MACR have Henry's law constants of 6.5 M/atm and 41 M/atm. Under common fog or cloud conditions, their fraction in the aqueous phase is $\ll 0.1\%$ - even without salting-out effects.

**This paragraph is effectively confusing and has been rewritten as following:**

**"In the present study, the CLEPS mechanism is extended to the oxidation of C1-4 precursors and follows the protocol described in detail in sections 3 and 4. Although isoprene is not significantly dissolved in the atmospheric aqueous phase, its oxidation products are considered (methylgyoxal-MGLY, glyoxal-GLY, acrolein-ACR, methacrolein-MACR, methylvinylketone-MVK) and are transferred into the aqueous phase depending on their solubility and reactivity in the aqueous phase."**

**Moreover, even MVK and MACR have low Henry's law constants, field measurements have reported aqueous concentrations 100 to 1000 times higher than the one estimated by the Henry's law constants (van Pinxteren et al., 2005).**

*van Pinxteren, D., Plewka, A., Hofmann, D., Müller, K., Kramberger, H., Svrcina, B., Bächmann, K., Jaeschke, W., Mertes, S., Collett Jr, J. L., and Herrmann, H.: Schmücke hill cap cloud and valley stations aerosol characterisation during FEBUKO (II): Organic compounds, Atmospheric Environment, 39, 4305-4320, 10.1016/j.atmosenv.2005.02.014, 2005.*

- p. 3, l. 31: I do not understand this. Even if all dissociated and hydrated forms are considered, I do not see how 87 chemical species yield 657 chemical forms. How can each species occur in about eight different forms (657 / 87).

**Yes, we agree with the comment, this is not explained in the text. For each organic species, we describe their hydration equilibria (mono, di and tri-hydrated forms) and their acid dissociation equilibria (mono and di-anion forms). The mechanism is available in the SI for more details. For example, for the organic compound 3-oxopyruvic acid CO(OH)COCHO (see table-mechanism-C3.pdf in the SI, page 29), we consider its acid dissociation equilibria but also the mono and di-hydrated forms of the acid form and of the mono-anion. This lead to the consideration of 8 different forms of the species 3-oxopyruvic acid. This compound then lead by oxidation to the formation of one major $RO_2$ (in this specific case) that is also in equilibrium with various chemical forms (8 forms). So for this compound, 16 different chemical forms have to be considered to describe its reactivity in CLEPS.**

**The text has been modified as follows: "This empirical reduction scheme helps to limit the number of species and reactions (657 different chemical forms (*i.e.*, hydrates, anions and derived radicals) representing 87 stable species reacting in 673 oxidation reactions)."**

- p. 6, l. 30; and p. 8, l. 32: Using Bond Dissociation Energies (BDEs) does not rely on estimates, These Energies have been measured and are tabulated [Benson et al., 1968; Benson, 1976].

**Yes, we agree with this comment. The sentence p. 6 l. 30 is modified as following: "In previous mechanisms, the most labile H-atom was identified (e.g., using Bond Dissociation Energy measurements)". The sentence p. 8, l. 32 is now in section 5 when we describe the differences between CLEPS and CAPRAM: "they assume that the reaction will proceed through the identified most probable pathway using the bond dissociation energy measurements".**

- p. 13, l. 8-11: This new text should be rewritten: 1) The first part sounds odd. I suggest changing it to 'We made sure that all species have am equivalent in the respective other phase, even if this species in that phase is not reactive.' 2) The mass transfer coefficients do not determine ultimately where a species resides as they describe the kinetics of the uptake. Only the Henry's law constant describes the thermodynamics and therefore where a species will reside.

**The text has been rewritten following the reviewer's comments.**

- p. 14, l. 12: 'Kinetic parameterization' of mass transfer or including chemical processes?

**We agree, we write "The kinetic parameterization of mass transfer in our cloud chemistry model".**

- p. 14, l. 30: The mass transfer is not represented by a rate constant as this implies a chemical transformation. I suggest using 'The mass transfer coefficients…'

**We agree, the text has been modified according to the comment.**

- p. 14, l. 31: Not all Henry's law constants are estimated.

**Yes we agree. We suppress the word "estimated" in the sentence.**

- p. 15, l. 9: Specify whether dry or wet deposition or both.

**We only consider dry deposition. This has been corrected.**

- p. 15, l. 18: Microphysics is not necessarily needed in order to simulate a more realistic cloud. In past models, reasonable approaches were taken by just switching on/off a cloud that is then present with constant LWC, drop size etc. for a shorter period than 12 hours.

**We agree with this. The sentence has been modified as follows "Future studies will use variable environmental conditions that require the consideration of microphysical processes with our multiphase chemical module."**

- p. 16, l. 9, and p. 18, l. 22: Replace 'dry' by 'cloud-free'. Real dry conditions (RH = 0%) are barely encountered in the atmosphere. What is the RH (water vapor concentration) during the gas-phase only runs?

**We agree with this. This has been corrected. The RH (10 %) conditions were precised in the text.**

- p. 16, l. 30/31 and 34/35: These lines are repetitive and can be combined.

**We agree with the reviewer. We combine these lines in one sentence: "Glyoxal, glycolaldehyde, pyruvic acid, glyoxylic acid and glycolic acid are readily soluble species that react in the aqueous phase (Herrmann et al., 2015), explaining the sharp decrease of their gas phase mixing ratios".**

- p. 16, l. 39: How is oxalic acid partitioning treated in the model? Oxalic acid has a fairly high vapor pressure. The only reason it remains in the aqueous/particle phase is the fact that is forms salts and stable complexes [Furukawa and Takahashi, 2011; Paris and Desboeufs, 2013].

**Oxalic acid partitioning is treated like other species in the model. A Henry's law constant is documented (Saxena and Hildemann, 1996). The complexation of oxalate ions with iron is also included in the mechanism (see R349-354 in the mechanism), and explains why oxalic acid largely remains in the aqueous phqse.**

*Saxena, P., and Hildemann, L. M.: Water-soluble organics in atmospheric particles: A critical review of the literature and application of thermodynamics to identify candidate compounds, J. Atmos. Chem., 24, 57-109, 1996.*

- p. 16, l. 39: What is meant by 'oxidation sink for all species'? A reduction in the oxidation rates because DOC is added and therefore OH is reduced? Clarify.

**We replaced the sentence "The addition of the missing aqueous oxidation sink for all dissolved species (red lines in Figures 3a & 3b) leads to higher concentrations of species for which reactive uptake is an overall sink (*e.g.*, glyoxal, glycolaldehyde) because the reduced aqueous HO$^\bullet$ concentrations (see Figure 4) limit the impact of the aqueous sink." with "The addition of the missing aqueous HO$^\bullet$ sink due to reaction with all dissolved unreactive species (red lines in Figures 3a & 3b) leads to higher concentrations of species for**

**which reactive uptake is an overall sink (*e.g.,* glyoxal, glycolaldehyde). In this additional HO˙ sink, the reduced aqueous HO˙ concentrations (see Figure 4) limit the impact of the aqueous sink.”**

- p. 17, l. 1-12, and l. 26-29: These lines can be combined as they are repetitive.

**To avoid the repetition, we replace: “The most important overall sink of HO˙ radicals in the aqueous phase is the reaction with the added DOC (64%), which results in a slight decrease in the simulated aqueous HO˙ concentrations in Figure 4. Besides DOC, simulated reactive organics are the most important HO˙ sinks, with C2 compounds contributing to 18%, and C4 compounds contributing to 12% of HO˙ destruction. C1 and C3 together are responsible for 5% of the HO˙ sink.” by “Organics are the most important HO˙ sinks, with DOC contributing to 64%, C2 compounds contributing to 18%, and C4 compounds contributing to 12% of HO˙ destruction. C1 and C3 together are responsible for 5% of the HO˙ sink.”**

- p. 17, l. 17: What is meant by 'pseudo steady state conditions'? Are they in thermodynamic equilibrium with their gas phase counterparts?

**Steady state conditions refer to conditions where sources are equal to sinks. We can remove the “Pseudo” from this expression to remove unneeded complications. In that case, the concentration of a given species is C=sources/sinks. However the explanation in this part is erroneous. We replace the sentences:**
**“Conversely, the organic species, which are mostly produced in the aqueous phase (formic, pyruvic, glyoxylic, and oxalic acid) have reduced sources and sinks when HO˙ radicals are scavenged by the added DOC. Because sinks and sources of acids due to HO˙ radicals should vary in equal proportions, the decrease in organic acids concentrations cannot be ascribed to reactivity with HO˙ radicals. We therefore have to consider fixed sinks that do not depend on HO˙ concentrations, *i.e.* photolysis and phase transfer. If we consider that acids reach pseudo steady state concentrations, we can assume that because photolysis and phase transfer are not modified by the additional DOC, some acids concentrations could decrease following their overall sources/sinks ratio.” with the following:**
**“Conversely, the organic species, which are mostly produced in the aqueous phase (formic, pyruvic, glyoxylic, and oxalic acid) have reduced sources and sinks when HO˙ radicals are scavenged by the added DOC. Their chemistry is slowed down and their rates of production are slower, giving lower maximum concentrations.”**

- p. 18, l. 18: This statement is not true. The solubility of small functionalized organics often exceeds that one of their larger homologues (e.g. $K_H$(glyoxal) = $10^5$ M/atm *vs* $K_H$(methylglyoxal) = 3000 M/atm).

**The sentence “Large molecules with high functionalization are statistically more soluble than smaller molecules” has been rephrased: “Large molecules with high functionalization are statistically more soluble than smaller, less functionalized molecules”.**

- p. 18, l. 36: Earlier it is said that photolysis processes are largely omitted due to the lack of data.

**Yes, we agree with the reviewer. We suppress “their reactivity by direct photolysis” from the sentence.**

- p. 19, l. 14/15: This text is confusing since it reads as if glyoxal *etc* are considered high molecular weight compound precursors.

**This sentence was suppressed in the final version of the manuscript. Because, the text was confusing.**

5   - p. 19, l. 17-19: This is not a conclusion of the current study.

**Yes, we agree. These sentences have been deleted from our manuscript.**

- p. 19, l. 38: Why are the mechanisms more useful for longer experiments? If all reactions are implemented correctly the time period or the number of phases (multiphase vs bulk aqueous phase) should not matter.

10  **Yes, we agree. The sentence was rewritten as follows: "The mechanisms are more likely to be useful for experiments involving multi-phases such as in environmental cloud chambers (see for example Brégonzio-Rozier et al., 2016)."**

**Technical comments**

- p. 1, l. 27: hydrosoluble should be replaced by water-soluble as it is used like that later on

**Done.**

5    - p. 1, l. 39: 'is' --> 'are'

**Done.**

- p. 3, l. 21: charges missing on $SO_4^{.}$ and $Cl_2^{.}$

**Done.**

- p. 6, l. 3: replace 'from' by 'by'

10    **Done.**

- p. 6, l. 6: Add 'constant' ('rate constant').

**Done.**

- p. 8, l. 30; p. 9, l. 5: reaction rates --> rate constants

**Done.**

15    - p. 15, l. 5: 'The simulation is located…' sounds odd. Reword.

**Corrected.**

- p. 15, l. 35: 'oxidants' --> 'oxidant'

**Done.**

- p. 16, l. 17 and 19: 'They emit...' and 'We emit…' is very colloquial. Reword.

20    **The sentences have been rewritten.**

- p. 16, l. 21: hydrocarbons --> hydrocarbon

**Done.**

- p. 17, l. 1: 'sinks' --> 'sink'

**Done.**

25    - p. 17, l. 16: 'has' --> 'have'

**Done.**

- p. 17, l. 37: Add 'reaction' (reactive sink reaction)

**Done.**

[revised manuscript text omitted]
. O/C ratios are calculated by counting the number of organic oxygen atoms in a givengas and cloud phases, and by dividing it by the number of organic carbon atoms in the same phase. The O/C ratios and $n_C$ are a measure of the extent to which long-chain organic species are oxidized and are therefore indicators of their functionalization and/or fragmentation. The O/C ratios and $n_C$ are a measure of the extent to which long-chain organic species are oxidized and can therefore be a proxy for their functionalization. One hour after the start of aqueous phase chemistrybeginning of the cloud, O/C in the aqueous phase has remained around 1.0 and $n_C$ has decreased to 2.8 after a sharp initial increase to 2.9, thus showing that fragmentation is a major process. O/C in the gas phase has increased from 0.74 to 0.77 and $n_C$ nC has decreased from 2.43 to 2.25. This result is in good agreement with other aqueous phase studies (Bregonzio-Rozier et al., 2016; Epstein and Nizkorodov, 2012; Epstein et al., 2013) and other models (Schrödner et al., 2014), but are in disagreement with field studies, probably due to a lack of descriptions of high molecular weight substances, and of their reactivity, as well as oligomerization processes. The higher O/C ratios obtained by Schrödner et al. (2014) after cloud event (1.8 for their rural case) can be due to important oxalic acid concentrations dissolved into the aqueous phase in their model, when the cloud is being formed. These values can be compared to the simulated O/C ratios in Schrödner et al.

(2014). They obtain much higher O/C ratios in clousafter cloud event (1.8 for their rural case). The main reason that could explain this difference is the exclusion of the counting of $CO_2$ (O/C = 2) and of carbonate ions (O/C = 3) in our calculation. Also in Schrödner et al. (2014), the uptake of some soluble gas phase species (e.g. formic and acetic acid) can explain greater O/C ratio. It is difficult to ascribe this difference to a specific reason. This could be due to an organic content more dominated by formic acid (O/C = 2). As no detail is given in Schrodner et al. (2014) about the O/C calculation, it could also be that they count $CO_2$ (O/C = 2) and carbonate ions (O/C = 3) in their calculation, which mechanically increases the resulting value. At the end of our simulation, Tthe 
[revised manuscript text omitted]